# Isobaric crosslinking mass spectrometry technology for studying conformational and structural changes in proteins and complexes

Jie Luo, Jeff Ranish*

Institute for Systems Biology, Seattle, United States

## eLife Assessment

This article presents a **valuable** new quantitative crosslinking mass spectrometry approach using novel isobaric crosslinkers. The data are **solid** and the method has potential for a broad application in structural biology if more isobaric crosslinking channels are available and the quantitative information of the approach is exploited in more depth.

**Abstract** Dynamic conformational and structural changes in proteins and protein complexes play a central and ubiquitous role in the regulation of protein function, yet it is very challenging to study these changes, especially for large protein complexes, under physiological conditions. Here, we introduce a novel isobaric crosslinker, Qlinker, for studying conformational and structural changes in proteins and protein complexes using quantitative crosslinking mass spectrometry. Qlinkers are small and simple, amine-reactive molecules with an optimal extended distance of ~10 Å, which use MS2 reporter ions for relative quantification of Qlinker-modified peptides derived from different samples. We synthesized the 2-plex Q2linker and showed that the Q2linker can provide quantitative crosslinking data that pinpoints key conformational and structural changes in biosensors, binary and ternary complexes composed of the general transcription factors TBP, TFIIA, and TFIIB, and RNA polymerase II complexes.

## Introduction

Proteins play a central role in the regulation of most biological processes by interacting with other molecules in complexes and interaction networks. Their ability to interact with other molecules to regulate biological processes depends on their structures. Protein tertiary structure is determined by and can be predicted from its primary amino acid sequence (**Jumper et al., 2021**). However, proteins, and protein complexes, are not static entities; they can assume multiple conformations. Conformational changes can be induced by post-translational modifications or through interactions with other molecules, and they allow proteins to execute their functions in a condition-dependent manner (**Sannigrahi et al., 2020**; **Sicoli et al., 2019**; **Khan and Kumar, 2009**; **Darling and Uversky, 2018**). Many disease-causing mutations impact protein function by altering the conformational landscape of a protein or protein complex (**Mashtalir et al., 2020**; **Prabantu et al., 2020**; **Carrell and Gooptu, 1998**). Experimental determination of protein conformational states is crucial in many cases to understand the mechanism of protein function and how function is altered in disease. There are two strategies, direct and indirect, for studying protein conformational changes. The direct strategy involves using methods such as X-ray crystallography, nuclear magnetic resonance , small angle X-ray

*For correspondence:
jranish@systemsbiology.org

Competing interest: The authors declare that no competing interests exist.

scattering, or cryo-EM to directly observe/compare structural changes under different conditions (*Vos et al., 2018*; *Catterall et al., 2020*; *Pilla et al., 2015*; *Chen et al., 2022*; *Burger et al., 2016*). Aside from the difficulties and limitations associated with each method, additional challenges for studying conformational changes include the fact that not all of the conformations can be easily crystallized or resolved, and many physiological conditions are not compatible with these methods. The indirect strategies are not able to directly observe the structural changes but can be used to deduce/infer structural changes based on changes in the output of the approach. These methods include limited proteolysis, immunochemical assays, fluorescence resonance energy transfer, chemical footprinting, and chemical crosslinking, some of which can be performed on proteins/complexes under physiological conditions (*Nadeau and Carlson, 2012*; *Sengupta and Udgaonkar, 2019*; *Rinas et al., 2016*; *Wang and Chance, 2017*). The main challenge for the indirect strategies is interpreting the data to infer the correct conformational/structural change out of numerous possibilities. Due to the limitations of direct and indirect approaches, the field of protein structural dynamics greatly benefits by combining these strategies with computational modeling (*Sala et al., 2022*; *Patel et al., 2018*; *Akey et al., 2022*).

Several mass spectrometry (MS)-based methods have been developed as indirect strategies to study conformational and structural changes in proteins and complexes. Hydrogen-deuterium exchange MS monitors the isotopic exchange rate between amide hydrogens along the protein backbone and the surrounding solvent to provide information about the folded state of the protein or complex (*Liu et al., 2020*). Protein painting-MS distinguishes solvent accessible regions from inaccessible regions, by using chemical dyes which non-covalently bind to solvent accessible regions, protecting them from trypsin digestion (*Haymond et al., 2019*; *Luchini et al., 2014*). Protein footprinting uses covalent chemical modifications and MS to identify surface exposed regions. Modification methods include dimethylation of lysine residues by formaldehyde and reducing reagent (*Bamberger et al., 2021*), glycine ethyl ester /EDC modification of carboxyl groups on aspartic and glutamic acid (*Liu et al., 2014*), NHS-based modification of lysine residues using isobaric Tandem Mass Tag (TMT) labeling reagents (*Zhou and Vachet, 2013*), isotopic succinic anhydride modification of lysine residues, and isotopic N-ethylmaleimide modification of cysteines (*Kahsai et al., 2011*). Hydroxyl radical protein footprinting, especially the fast photochemical oxidation of proteins, coupled with MS has greatly advanced to permit the study of surface exposed regions of proteins in cell lysates and in vivo (*Hambly and Gross, 2005*; *Espino et al., 2015*; *McKenzie-Coe et al., 2022*). All of these MS-based methods involve reactions with solvent-accessible residues or surfaces, and are suitable for studying structural changes in proteins and complexes. However, since these methods do not provide information about spatial proximities between residues/domains, information about the relative locations of residues/domains in different conformational states is limited, especially for larger assemblies. Also, a conformational change does not necessarily involve a change in solvent accessibility.

Crosslinking-mass spectrometry (CLMS or CXMS) has been widely used to provide distance restraints between residues and relative positioning of domains in large protein complexes (*Chen et al., 2010*; *Chavez et al., 2015*; *Mashtalir et al., 2018*; *Abdella et al., 2021*). Importantly, when performed in a quantitative manner, CLMS can provide information about conformational changes in proteins and complexes, as well as changes in protein–protein interactions. Multiple quantitative crosslinking-MS (qCLMS) strategies have been developed to study conformational changes. TMT labeling of crosslinked samples has been used to quantify crosslinked and normal peptides (*Yu et al., 2016*; *Ruwolt et al., 2022*). In this approach, two or more crosslinked samples are individually digested with trypsin and then labeled with TMT reagents prior to combining the samples. The multiple steps prior to combining the samples can result in quantification artifacts that are due to sample processing differences rather than to conformational or interaction changes, thus complicating data interpretation. To avoid these issues, qCLMS strategies that involve co-digestion of crosslinked samples have been developed. These approaches employ isotopically labeled crosslinkers such as d0/d4 bis[sulfosuccinimidyl] suberate (BS3) (*Chen and Rappsilber, 2019*; *Mendes et al., 2019*), d0/d4 *bis*(sulfosuccinimidyl)-glutarate, -pimelate, and -sebacate (*Müller et al., 2001*), d0/d12 ethylene glycol *bis*(succinimidylsuccinate) (*Petrotchenko et al., 2005*), d0/d12 MS cleavable DSBU (*Ihling et al., 2020*), d0/d8 MS cleavable CBDPS (*Makepeace et al., 2020*), or SILAC-labeled samples (*Chavez et al., 2015*). Relative quantification is based on the MS1 intensities of the isotopically heavy and light crosslinked peptides after identification of the crosslinked peptides. However, like all MS

strategies that are based on isotopically heavy- and light-labeled peptides, the differentially labeled samples double or triple the MS1 complexity, which can decrease sensitivity and reproducibility; many identified crosslinked peptides are difficult to confidently quantify because they are of low abundance and difficult to distinguish from the background noise signals; and multiple or differential database searches result in extra difficulties for confident light/heavy crosslinked peptide identification and quantification. Based on their MS-cleavable protein–interaction reporter (PIR) design, Bruce and colleagues have developed 2-plex and 6-plex isobaric quantitative PIRs (iqPIRs) with limited sample handling prior to MS analysis, a single MS1 spectrum for each crosslinker-modified peptide and quantification based on MS2 reporter ions (*Chavez et al., 2020*; *Chavez et al., 2021*). However, PIR crosslinkers are bulky molecules that are more suitable for studying dynamic protein–protein interactions rather than conformational and structural changes. In fact, many of the aforementioned crosslinker designs are too complicated for routine studies of conformational and structural changes in proteins and complexes (*Wippel et al., 2022*). Here, we report a novel, simple, and small, isobaric crosslinker design for studying conformational and structural changes. This new crosslinker, we termed Qlinker, has an optimal spacer arm length (~10 Å) for crosslinking-based studies of protein conformational and structural changes, and uses 1-imino-2.6-dimethylpiperidin-1-ium reporter ions (similar to TMT reporter ions; *Thompson et al., 2003*) for quantification. We have used the 2-plex Q2linker to study conformational changes in biosensors, binary and ternary complexes composed of the general transcription factors TATA box binding protein (TBP), TFIIA, and TFIIB, and structural rearrangements that accompany the transition of RNA polymerase II (pol II) from a 10 subunit core complex to a 12 subunit holoenzyme. In each of these studies, the Qlinker approach provided quantitative crosslinking data that pinpointed key conformational and structural transitions in the proteins/complexes. We expect that this novel strategy for studying conformational and structural changes in proteins and complexes will be easily adopted for routine use by the community.

## Results

### Design of isobaric Qlinkers

We sought to develop a simple isobaric CLMS-based strategy, called Qlinker, that permits probing conformational changes in proteins/protein complexes by reliably quantifying the relative abundances of crosslinker-modified peptides derived from samples of the protein/complex in different structural states. An important consideration when designing isobaric amine-reactive crosslinkers is that the m/z's of the reporter ions do not overlap the m/z's of other fragment ions, which could skew quantification. Unlike iTRAQ or TMT modified peptides, which cannot generate unmodified $b1^+$ ions (because all peptide N-termini are labeled), the Qlinker-modified peptides, which are generated by enzymatic digestion after the crosslinking reaction is complete, have free N-terminal amines and will generate unmodified $b1^+$ ions. (A $b1^+$ ion is an ion with a charge state of +1 corresponding to the first N-terminal amino acid residue after breakage of the first peptide bond.) Since isoleucine and leucine residues generate $b1^+$ ions at 114.09 m/z and asparagine and aspartate residues generate $b1^+$ ions at 115.05 m/z and 116.03 m/z, respectively, we sought to incorporate a moiety into the Qlinker that generates reporter ions with m/z's that do not overlap with the m/z's of $b1^+$ and immonium ions. Thus, we decided to use 1-imino-2.6-dimethylpiperidin-1-ium as the reporter ion moiety, which is the same as the moiety used in 126–134 TMT reagents (*Figure 1—figure supplement 1*). We searched 83,242 mass spectra obtained from the analysis of a BS3 crosslinked sample of the 1 MDa yeast Mediator complex and only found 141 spectra (0.17%) containing 126.128 ions and 18 spectra (0.02%) containing 127.131 ions within a 40 ppm mass tolerance. These results indicated that the 126.128 and 127.131 reporter ions can be used for accurate quantification of Qlinker-modified peptides with little interference from fragment ions derived from peptides that are not modified with Qlinkers.

Another important consideration in the design of the amine-reactive Qlinker is that it can react with ε-amines of lysine that are in close proximity. Iminodiacetic acid, iminodipropionic acid, and iminodibutyric acid were considered as base structures. Due to the flexibility of lysine side chains, it was found that Cα-Cα distances of residues crosslinked with amine-reactive crosslinkers are more useful as distance restraints than the distances between crosslinked ε-amines. The BS3 crosslinker has a spacer arm of 11.4 Å when fully extended and can crosslink lysine residues whose Cα atoms are up to 30 Å apart (*Merkley et al., 2014*). The zero-length crosslinker EDC/DMTMM can crosslink a lysine

residue and an aspartate or a glutamate residue with Cα atoms up to 20 Å apart (*Leitner et al., 2014*). We reasoned that crosslinkers based on iminodiacetic acid or iminodipropionic acid with extended crosslinker spacer arms of 7.6 Å and 10.1 Å, respectively, would provide similar structural information, while the backbone of an iminodiacetic acid-based reagent would be more restricted than that of an iminopropionic acid-based reagent. At the same time, iminodibutyric acid quickly generates a dark red color during activation (probably due to lactone formation). We decided to use iminodipropionic acid as the base structure for the new quantitative Qlinker (*Figure 1—figure supplement 1*).

## Experimental procedure and Q2linker quantification

The two isobaric Q2linkers are named C1q2 and C2q2, synthesized from 1-$^{13}$C and 2-$^{13}$C bromoacetic acid (BAA) respectively (*Figure 1a*, *Figure 1—figure supplement 1*, and 'Materials and methods'). A typical experiment uses equal amounts of C1q2 and C2q2 to crosslink equal amounts of proteins or protein complexes in different structural states. After quenching the reactions with ammonium sulfate, the two samples are combined and mixed well, reduced and denatured, trypsin digested, and analyzed by MS using a stepped HCD MS2 method (*Diedrich et al., 2013*). The Qlinker-modified peptide spectra are identified as either monolinks or crosslinks by different database search engines. The ion signals from the 126.1277 (C1q2) and 127.1311 (C2q2) reporter ions are then extracted from identified spectra to compute the relative abundance of the modified peptides in the different samples (*Figure 1a*).

To evaluate the ability of Q2linkers to provide accurate quantification of crosslinker-modified peptides, we crosslinked equal amounts of affinity-purified RNA polymerase I (pol I) with either C1q2 or C2q2, digested the two samples separately, and then combined the resulting peptides at known ratios for MS analysis (*Figure 1—figure supplement 2*). Examples of a monolinked peptide spectrum (*Figure 1b*) and a crosslinked spectrum (*Figure 1c*) are shown, with the reporter ion intensities from the indicated mixing ratios. The 126 and 127 reporter ion intensities were extracted from the identified spectra corresponding to 298 monolinks and 32 crosslinks, within a mass tolerance of 0.005 Da (~40 ppm) and adjusted by correction factors calculated from the isotopic distribution ('Materials and methods'). The measured log2 ratios of 127/126 intensity were then compared with the expected log2 ratios. The measured ratios from both monolink spectra (*Figure 1d*) and crosslink spectra (*Figure 1e*) were strongly correlated with the expected ratios with slopes close to 1 (*Figure 1—figure supplement 3*). At each mixing ratio, the average measured ratio was slightly lower than the expected ratio (intercept ~–0.4), with a standard deviation around ±0.25, suggesting that there were slightly more crosslinker-modified peptides in the C1q2 sample prior to mixing. This may be due to some peptide loss in the C2q2 sample during trypsin digestion and C18 preparation prior to mixing, or there was slightly less C2q2 crosslinker in the reaction. This also highlights the importance of combining crosslinked samples at an early stage to minimize sample handling variations. The standard deviation is larger for the samples mixed at 5:1 and 10:1 C1q2 labeled (126) to C2q2 labeled (127) (around ±0.42 and ±0.78, respectively). We think this could be due to either the higher natural isotope contribution (~10%) from the 126 reporter ion to the 127 reporter ion, or interference from other fragment ions. Overall, these experiments demonstrate that the Q2 quantification strategy can accurately reflect the relative abundance of Q2linker-modified peptides.

We then affinity-purified pol II from a yeast strain expressing a FLAG-tagged version of pol II subunit Rpb3 and crosslinked equal amounts of the 12 subunit pol II complex with Q2linkers in the presence and absence of α-amanitin. No major structural rearrangements occur in pol II upon α-amanitin binding (1i3q.pdb for free pol II and 1k83.pdb and 3cqz.pdb for α-amanitin bound pol II). The distributions of log2(127/126) ratios for monolinked (total 174), intralinked (total 154), and interlinked (total 77) peptides were all centered around 0 with no significant differences. Also, 98% of the log2 ratios fell within the range ±0.5 (*Figure 1—figure supplement 4*). This experiment demonstrates that Q2linkers provide reliable crosslink and monolink quantification with no abnormal ratios when applied to a protein complex in a situation where no significant conformational changes and/or structural rearrangements are known to occur. We initially designed this experiment thinking that α-amanitin binding to pol II might affect Q2linker modification of a lysine residue(s) near the α-amanitin binding site, and we might be able to identify the α-amanitin binding site by analysis of quantitative Q2linker MS data. Unfortunately, no Q2linker-modified peptides were identified near the site where α-amanitin binds. This experiment also highlights one of the limitations of residue-specific, quantitative CLMS

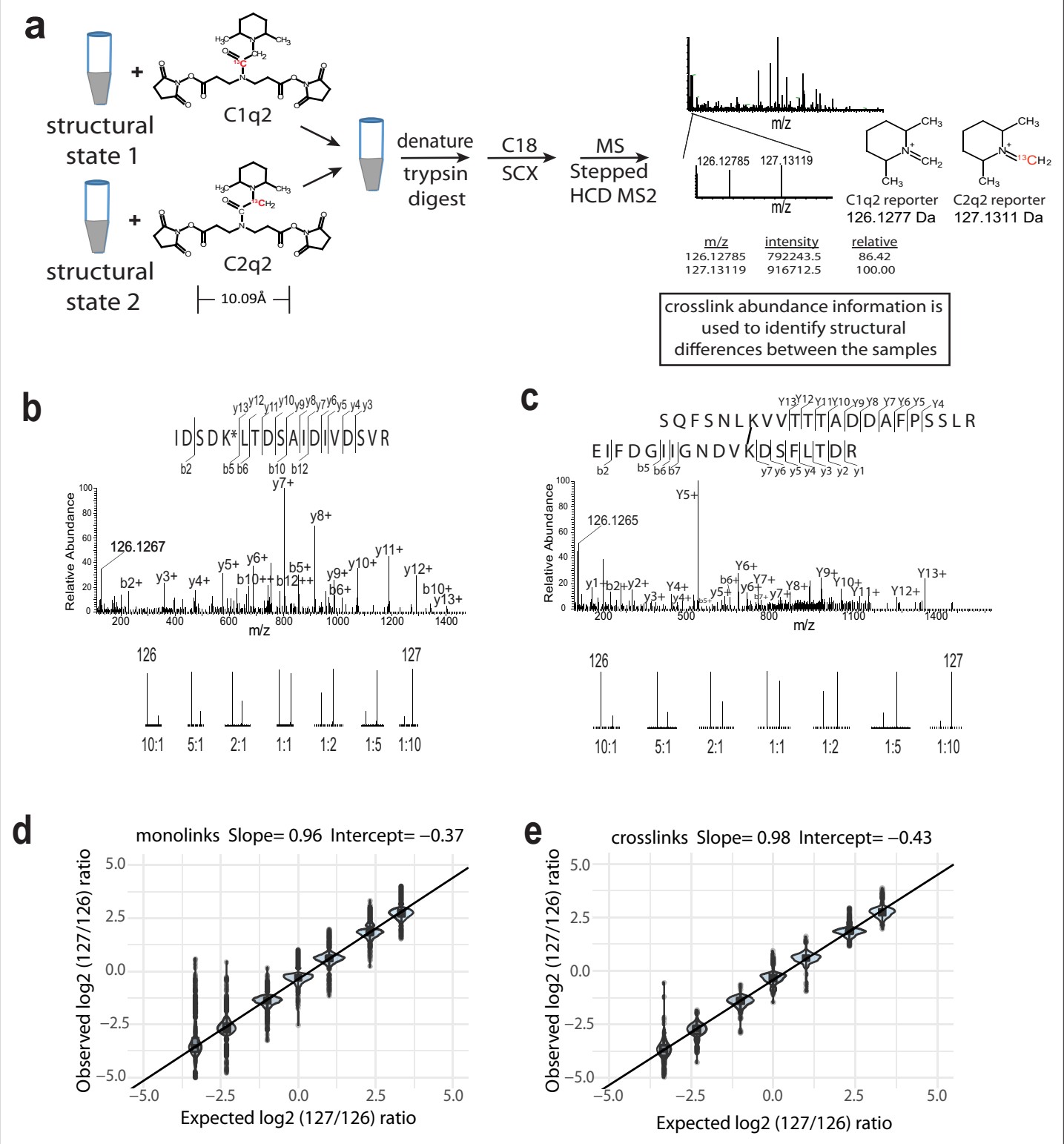

**Figure 1.** The isobaric Qlinker quantitative crosslinking-mass spectrometry (CLMS) approach for studying conformational and structural changes in proteins and protein complexes. (**a**) The structure of Q2linkers and the general scheme for qCLMS using Q2linkers. The $^{13}$C atom in C1q2 or C2q2 is indicated by red font. (**b–e**) Experiment to evaluate the ability of Q2linkers to quantify the relative abundances of crosslinks and monolinks derived from Q2linker modification of TAP-tag affinity-purified pol I at designated ratios (~20 ug for each mixing ratio). Example spectra of a monolinked peptide (**b**) and a crosslinked peptide (**c**) with the observed reporter ion intensities at each mixing ratio. (**d**) Observed vs. expected log2 (127/126) reporter ion

*Figure 1 continued on next page*

*Figure 1 continued*

ratios for monolinks. (**e**) Observed vs. expected log2 (127/126) reporter ion ratios for crosslinks. Boxplot in R was used to create the graphs in (**d**) and (**e**). Center line, median; box limits, upper and lower quartiles; whiskers, 1.5× interquartile range; points, outliers. The experiment was performed once.

The online version of this article includes the following source data and figure supplement(s) for figure 1:

**Source data 1.** Table containing information about the monolinks reported in *Figure 1b and d*.

**Source data 2.** Table containing information about the intralinks reported in *Figure 1c and e*.

**Source data 3.** Table containing information about the interlinks reported in *Figure 1c and e*.

**Figure supplement 1.** Synthesis of Q2linkers.

**Figure supplement 2.** Experimental procedure to evaluate the ability of Q2linkers to quantify the relative abundances of crosslinks and monolinks derived from Q2linker modification of affinity-purified pol I.

**Figure supplement 2—source data 1.** File containing the original gel for *Figure 1—figure supplement 2*, indicating the relevant bands and molecular weight markers.

**Figure supplement 2—source data 2.** Original file for the gel displayed in *Figure 1—figure supplement 2*.

**Figure supplement 3.** Observed vs. expected log2 (126/127) reporter ion ratios for individual monolinks (**A**) and crosslinks (**B**) at different mixing ratios.

**Figure supplement 4.** The ratio distributions for interlinks, intralinks, and monolinks from the pol II +/-a-amanitin experiment.

methods in general. Reactive residues must be available near the region of interest, and the modified peptides must be identifiable by MS.

## The study of conformational changes in biosensors using Q2linkers

Protein biosensors are polypeptides that undergo conformational changes or switches upon receiving an input signal. We next designed experiments to see whether Q2linkers can detect conformational changes in biosensors. We tested two commercially available proteins: maltose binding protein (MBP) and calmodulin (CaM). MBP is a 370 amino acid polypeptide that buries the maltose ligand between a cleft formed by its two domains and exhibits a conformational change from an open ligand-free conformation (1mpb.pdb) and a closed maltose-bound conformation (1n3w.pdb). However, it is the 'balancing interface' on the opposite side of the ligand binding cleft that maintains the open conformation; upon maltose binding, this interface is disrupted and becomes more solvent exposed (*Telmer and Shilton, 2003*). To evaluate the ability of Q2linkers to detect conformational changes in MBP, we performed a crosslinker swapping experiment in which we crosslinked 10 ug of MBP in the presence and absence of 10 mM maltose with C1q2 and C2q2, respectively, in one experiment, and, in a second experiment, we crosslinked MBP in the presence and absence of maltose with C2q2 and C1q2, respectively. In each experiment, we combined the C1q2 and C2q2 crosslinked samples prior to trypsin digestion and C18 cleaning, and used 1 ug for MS analysis. In both experiments, we identified one monolinked peptide ($_{306}$SYEEELAK*DPR$_{316}$) that was fivefold more abundant in the samples containing maltose compared to the samples without maltose (*Figure 2a*), suggesting K313 is modified more readily by the Q2linkers when maltose is present. Interestingly, this peptide happens to reside within the 'balancing interface' of MBP. In the closed conformation, the sequence between 301 and 312 forms an alpha helix (*Figure 2a*, gray structure) and upon binding to maltose, the alpha helix relaxes and unwinds (*Figure 2a* cyan structure). However, K313 itself is solvent exposed in both structures, so the increase in K313 modification in the open conformation cannot be simply explained by an increase in accessibility upon maltose binding. Comparison of the open and closed conformations of MBP reveals that K313 can form a salt bridge with E310 in the open conformation (K313-NE to E310-OE2 distance of ~3.3 Å), and this salt bridge is broken in the closed conformation. It is likely that the salt bridge between K313 and E310 limits the NHS ester-based modification of K313 as revealed by the quantitative difference in the abundance of the Q2linker-modified peptide containing K313.

Calmodulin (CaM), a small polypeptide composed of 148 AAs, is a well-studied biosensor that undergoes conformational changes upon binding to calcium and CaM-binding peptides (CBPs) (*Meister and Joshi, 2013*). To evaluate the ability of Q2linkers to detect conformational changes in CaM, we crosslinked 25 ug CaM, in the presence or absence of 20 mM CaCl$_2$ and a nearly 1:1 molar ratio of CBP (8 ug) derived from alphaII-spectrin with C2q2 and C1q2, respectively. After crosslinking, the samples were combined, trypsin digested, and 1 ug of the sample was analyzed by MS. One crosslinked peptide was identified containing a Lys78 to Lys95 linkage, which was fivefold less abundant in

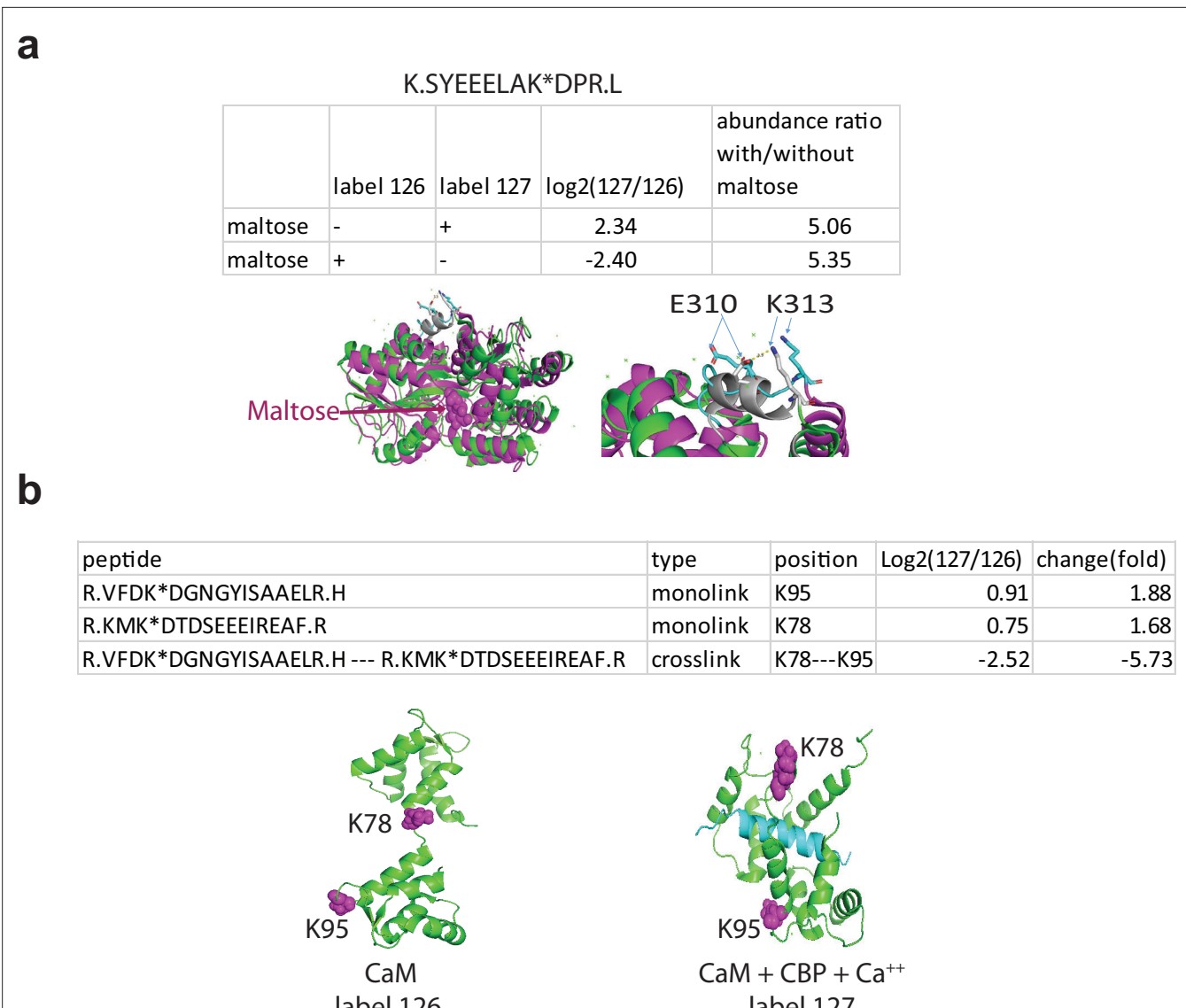

**a**

**K.SYEEELAK\*DPR.L**

| | label 126 | label 127 | log2(127/126) | abundance ratio with/without maltose |
|---|---|---|---|---|
| maltose | - | + | 2.34 | 5.06 |
| maltose | + | - | -2.40 | 5.35 |

**b**

| peptide | type | position | Log2(127/126) | change(fold) |
|---|---|---|---|---|
| R.VFDK\*DGNGYISAAELR.H | monolink | K95 | 0.91 | 1.88 |
| R.KMK\*DTDSEEEIREAF.R | monolink | K78 | 0.75 | 1.68 |
| R.VFDK\*DGNGYISAAELR.H --- R.KMK\*DTDSEEEIREAF.R | crosslink | K78---K95 | -2.52 | -5.73 |

**Figure 2.** Q2linkers detect conformational changes in protein biosensors. (**a**) Quantification of a monolinked peptide from maltose binding protein (MBP) with or without maltose. Structures of MBP in the open, ligand-free conformation (green, 1mpb.pdb) and the closed, maltose-bound conformation (brown, 1n3w.pdb) are shown on the left. A close-up view of the 'balancing interface' is shown on the right. In the closed conformation, the sequence between amino acids 301–312 forms an alpha helix (gray) and K313 forms a salt bridge with E310. The helix and the salt bridge are disrupted in the open conformation (cyan). A crosslinker swapping experiment was performed on the same preparation of MBP; a technical replicate. (**b**) Quantification of monolinked and crosslinked peptides involving K78 and K95 from apo-CaM and CaM with $Ca^{2+}$ and CBP. Structures of apo-CaM (left) and CaM + CBP (blue) + $Ca^{2+}$ (right) are shown with key lysine residues space filled and magenta. The experiment was performed once.

the presence of CBP and $CaCl_2$ (*Figure 2b*), suggesting that crosslinking between these two residues is greatly inhibited upon calcium and CBP association with CaM. We also identified the corresponding monolinked peptides containing these two residues and found that their abundances were slightly increased upon binding of calcium and CBP (*Figure 2b*). These results suggest that the reduced abundance of the crosslinked peptide containing the Lys78-Lys95 linkage cannot be explained by limited accessibility of these two sites in the presence of $Ca^{2+}$ and CBP. Indeed, the crystal structures of apo-CaM (1cfd) and CBP-bound-CaM (2bbm) show no changes in the surface exposure of Lys78 and Lys95. However, CBP binds to the region between Lys78 and Lys95 and interferes with the ability of the Q2linker to form a crosslink between them due to either steric hindrance or decreased flexibility

of CBP-bound CaM (*Figure 2b*). The results from the MBP and CaM qCLMS experiments show that the Qlinker approach can detect conformational changes in biosensors.

## Q2linkers detect structural changes during TFIIA/TBP/TFIIB ternary complex formation

When proteins interact with one another to form a quaternary protein complex, the complex often will have lower energy states with subtle conformational changes and/or structural rearrangements, albeit many of these changes are unknown and are difficult to detect. We next wanted to see whether Q2linker can be used to detect subtle conformational changes, as well as protein–protein interaction changes during quaternary complex formation. During pol II transcription initiation, general transcription factors, including TBP, TFIIA, and TFIIB, play central roles in promoter recognition, preinitiation complex (PIC) formation, and start site selection (*Murakami et al., 2015*; *Tsai and Sigler, 2000*). During PIC formation, TBP binds to TATA-containing promoter DNA, and TFIIA, composed of two subunits in yeast, TOA1 (or TFIIA1) and TOA2 (or TFIIA2) (*Ranish and Hahn, 1991*), associates with and stabilizes the TBP-TATA DNA complex (*Tan et al., 1996*; *Figure 3c*). The C-terminal core domain of TFIIB also interacts with TBP, while the N-terminal zinc ribbon domain and B finger of TFIIB reach into the active center of pol II (*Chen and Hahn, 2004*). The central linker region of TFIIB (81-211) snakes down the central cleft of pol II, interacting with the Rpb1 clamp and Rpb2 protrusion loops to stabilize the PIC complex (*Murakami et al., 2015*). TFIIA and TFIIB independently interact with the N- and C-terminal lobes of TBP, respectively; no interaction between TFIIA and TFIIB has been described (*Tan et al., 1996*; *Andel et al., 1999*). We incubated purified, recombinant TBP with either TFIIA or TFIIB (1:1 by weight for both reactions) to form TBP-TFIIA and TBP-TFIIB complexes, and then crosslinked each complex separately with C1q2 crosslinker. At the same time, we incubated TBP, TFIIA, and TFIIB (1:1:1 by weight) to form the TBP-TFIIA-TFIIB complex, and crosslinked the sample with C2q2 linker. After quenching the crosslinking reactions, all of the samples were combined, digested with trypsin, fractionated by strong cation exchange (SCX) chromatography, and prepared for MS analysis. SCX was used to reduce the complexity of the peptide mixtures. After database searching with both pLink2 and Nexus algorithms, 61 interlinks were identified. We grouped the interlinks between the different proteins in the complexes and found a tight distribution of log2 (127/126) reporter ion intensity ratios centered around 0 with most ratios between 1.4 to –1.4-fold (*Figure 3a*). Two crosslinks between TFIIA1 and TFIIB have large abundance differences (>4.0-fold) because crosslinks between these proteins can only be observed in the TBP-TFIIA-TFIIB samples crosslinked with C2q2 (127). TFIIA and TFIIB were not co-incubated in either of the samples crosslinked with the C1q2 (126) crosslinker. 202 intralinks were also identified. Since there is twice as much TBP in the pooled C1q2 crosslinking reactions, the average log2 (127/126) ratio for TBP intralinks is ~–0.5 (~1.4-fold more in the samples crosslinked with the C1q2 crosslinker). We think this is reasonable as the crosslinked peptide ratios are not only related to the protein abundance but also to crosslinking patterns, crosslinking efficiency, monolinks, mis-cleavages, etc. We re-centered the log2 (127/126) ratios for TBP intralinks at –0.5 while leaving the ratios for the other intralinks unchanged to compare the ratio distribution of the intralinks (*Figure 3b*). Like interlinks, most intralink ratios show no significant changes between the two experimental conditions examined here.

No significant conformational changes have been observed previously based on the crystal structures and biochemical studies of the TBP/TFIIA, TBP/TFIIB, and TBP/TFIIA/TFIIB complexes with TATA DNA (*Murakami et al., 2015*; *Tan et al., 1996*; *Nikolov et al., 1995*; *Dion and Coulombe, 2003*; *Kays and Schepartz, 2000*). This is due to the fact that TFIIA and TFIIB independently interact with the N- and C- lobes of TBP, respectively, and no direct interaction between TFIIA and TFIIB is observed in the cryo-EM structures (*Murakami et al., 2015*; *Andel et al., 1999*). In addition, only the core, conserved regions of TFIIB, TBP, and TFIIA were mapped in the crystal/cryo-EM structures. We used full-length versions of TFIIB, TBP and TFIIA in our study. Most of the interlinks and intralinks identified in our study have no significant abundance changes between the samples, in agreement with previous structural studies (*Figure 3a and b*). Interestingly, the largest interlink changes, besides the previously described crosslinks between TFIIA1 and TFIIB, involve crosslinks between the N-terminal region of TBP and the central linker region of TFIIB, both of which are unstructured and absent in the crystal structures. The largest ratio changes among the intralinks all involve the central linker region of TFIIB, whose structure has only been observed in complexes containing pol II (*Figure 3c*). Our

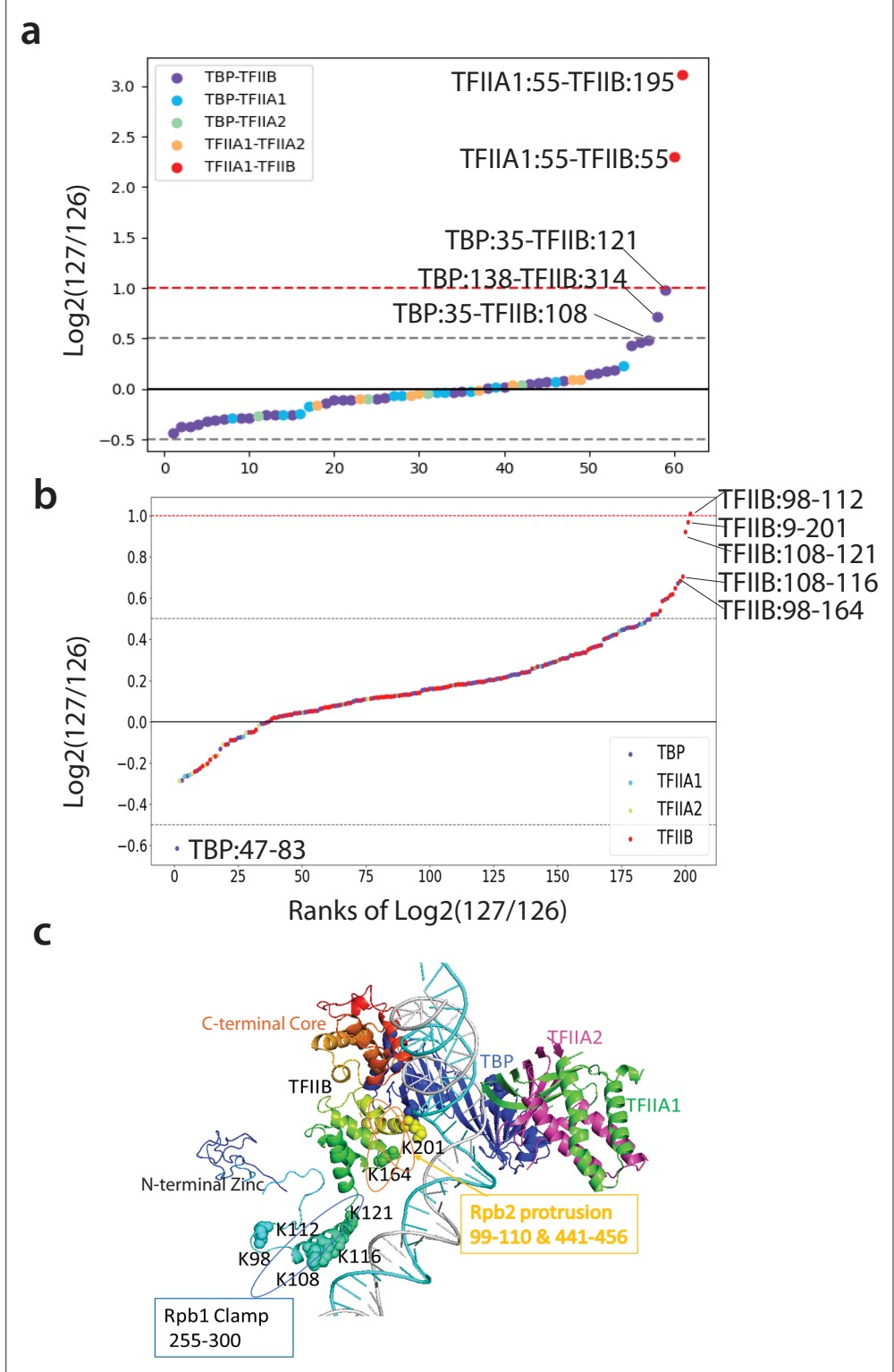

**Figure 3.** Q2linkers detect structural changes during TFIIA/TBP/TFIIB ternary complex formation. (**a, b**) The distribution of log2 (127/126) ratios for interlinks (**a**) and intralinks (**b**) from the Qlinker experiment comparing the TFIIA/TBP/TFIIB ternary complex (127) to the TFIIA/TBP and TFIIB/TBP binary complexes (126). The x-axis is the ranking of the log2 (127/126) ratios. Crosslinks with the highest ratios are labeled. (**c**) The structure of TFIIA, TBP, and TFIIB in the pol II PIC complex (5fmf) is shown. TFIIB is rainbow colored with blue at the N-terminus and red at the C-terminus. Residues in the central region of TFIIB that are involved in crosslinks with high ratios are shown

*Figure 3 continued on next page*

*Figure 3 continued*
as space filled. The regions of TFIIB that interact with the Rpb1 clamp and Rpb2 protrusion are also shown. The experiment was performed once.

The online version of this article includes the following source data for figure 3:

**Source data 1.** Table containing information about the interlinks reported in *Figure 3a*.

**Source data 2.** Table containing information about the intralinks reported in *Figure 3b*.

results suggest that the addition of TFIIA to the TBP/TFIIB complex induces a change in the conformation of the TFIIB linker region. This may involve the rearrangement of the TBP N-terminal domain (NTD). Previously, it was suggested that TFIIA stimulates TBP binding to DNA by causing a structural change involving the NTD of TBP, which alleviates an inhibitory function of the NTD (*Lee et al., 1992*; *Bleichenbacher et al., 2003*). This TFIIA induced change in the conformation of the TBP NTD may be reflected by the reduced abundance of the intralink between TBP residues 47 and 83 in the sample containing TBP/TFIIA/TFIIB (*Figure 3B*). Thus, it is possible that the interaction between TFIIA and TBP not only increases TBP's affinity for TATA DNA, but also causes a structural rearrangement in both TBP and TFIIB so that TFIIB may be better positioned to interact with pol II for PIC formation (*Chen and Hahn, 2004*; *Glossop et al., 2004*; *Bangur et al., 1999*). Our study has captured dynamic structural changes which accompany the formation of the TFIIA/TBP/TFIIB complex.

## Probing Rpb4/7-induced structural changes in pol II using Q2linkers

We next evaluated the ability of Q2linkers to detect conformational changes in large protein complexes. Yeast pol II is a >0.5 MDa protein complex and a good model system to study structural changes in large complexes as the 12-subunit holo-pol II assumes a conformation that is distinct from that of the 10-subunit, core pol II complex, lacking the Rpb4/Rpb7 dimer (*Cramer et al., 2001*; *Bushnell and Kornberg, 2003*; *Oberthuer et al., 2017*). In core pol II (1i3q), the 'clamp' is in an 'open' state, allowing formation of a straight channel for DNA template entry (*Cramer et al., 2001*). In the core pol II elongation complex (1nik) and the holo-pol II enzyme (5u5q), a massive movement of the 'clamp', which rotates by about $30^0$ with a maximum displacement >30 Å at external sites, results in the 'closed' state (*Bushnell and Kornberg, 2003*; *Gnatt et al., 2001*). However, most of the clamp moves as a rigid body and the large structural movement is produced by conformational changes in five 'switch' regions (*Gnatt et al., 2001*; *Figure 4—figure supplement 1*, magenta). The Rpb4/Rpb7 heterodimer binds to core pol II through interactions between Rpb6 (91–105) and Rpb1 (1440–1452) with Rpb7. *RPB4* is not required for viability while *RPB7* is essential for growth in yeast. We affinity-purified holo-pol II from a yeast strain expressing FLAG-tagged Rpb3, and we purified pol II lacking Rpb4 from an *RPB4* deletion strain expressing FLAG-tagged Rpb2 (*Figure 4a*). We crosslinked ΔRpb4 pol II and holo-pol II with C1q2 and C2q2, respectively, in experiment I, and then performed a crosslinker swapping experiment in which we crosslinked ΔRpb4 pol II and holo-pol II with C2q2 and C1q2, respectively, in experiment II (*Figure 4a*). The samples were analyzed by MS, and Comet database searching against the yeast proteome identified unmodified peptides and monolinks corresponding to 234 proteins (>99% probability). We used pLink2 (*Chen et al., 2019*) and Nexus (*Mashtalir et al., 2018*) to identify crosslinks by searching a database composed of the sequences of the 12 pol II subunits, and then extracted the ion intensities for the 126 and 127 reporter ions from the identified spectra for their relative quantification. We then averaged the log2(126/127) ratios from each spectrum corresponding to a Qlinker-modified site or pair of crosslinked sites (*Supplementary file 1*), and plotted the ratios from experiment I on the x-axis and experiment II on the y-axis of the graphs shown in *Figure 4b–d*. Each green dot represents one unique site or pair of sites identified in both experiments. Each blue or yellow dot represents a unique site or pair of sites identified only in experiment I or experiment II, respectively.

Here, 101 interlinks were identified in both experiments I and II, 47 interlinks were identified in experiment I only, and 116 interlinks were identified in experiment II only (*Figure 4b*). However, among all of the interlinks that were only identified in one of the experiments, only three in experiment I and four in experiment II exhibit a greater than two fold intensity difference. This suggests that the inability to identify most of these crosslinked peptides in both experiments is mainly due to undersampling during MS analysis of the complex samples, rather than the absence of the crosslinked peptides in one

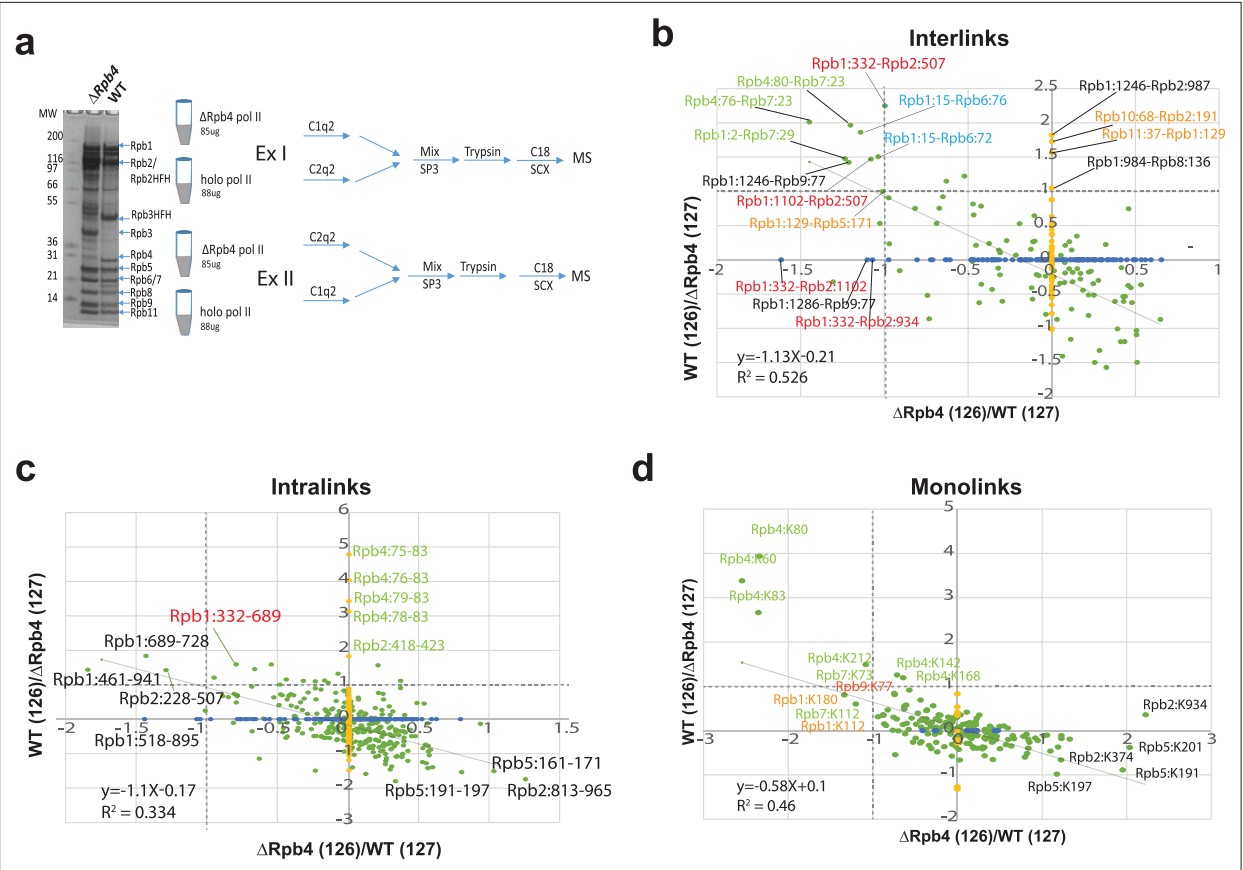

**Figure 4.** Q2linkers detect conformational changes in large protein complexes. (**a**) The SDS-PAGE gel of affinity-purified pol II from WT and ΔRpb4 strains and experimental design for studying conformational changes in RNA polymerase II due to deletion of Rpb4. (**b–d**) The log2(126/127) ratio comparisons in experiment I and II for holo-Pol II and ΔRpb4-pol II. x-axis is ΔRpb4-pol II (126)/holo-pol II (127). y-axis is holo-pol II (126)/ΔRpb4-pol II (127). Enrichment in the holo-pol II sample (labeled with C2q2 in experiment I) will have a smaller log2(126/127) ratio on the x-axis; enrichment in the holo-pol II (labeled with C1q2 in experiment II) will have a higher log2(126/127) ratio on the y-axis. Each green dot corresponds to one unique pair of crosslinking sites identified in both experiments. Each blue dot on the x-axis corresponds to one unique pair of crosslinking sites identified only in experiment I and each yellow dot on the y-axis corresponds to one unique pair of crosslinking sites identified only in experiment II. Only the green dots are used for the linear regression analysis. (**b**) Interlinks, (**c**) intralinks, and (**d**) monolinks. Dashed lines indicate log2 ratio = 1. The same preparation of each protein complex was used in a crosslinker swapping experiment; a technical replicate.

The online version of this article includes the following source data and figure supplement(s) for figure 4:

**Source data 1.** File containing the original gel for *Figure 4a*, indicating the relevant bands and molecular weight markers.

**Source data 2.** Original file for the gel displayed in *Figure 4a*.

**Figure supplement 1.** Structural comparison of holo-pol II (5u5q) and core-pol II (li3q).

of the experiments. This also highlights the importance of reliable methods for quantification in CLMS experiments involving complex samples, where undersampling of crosslinked peptides is exacerbated due to the increased complexity of the crosslinker-modified samples and the inefficiency of the cross-linking reaction. Considering all of the interlinks identified in both experiments, there is a rough anti-correlation between the ratios measured in experiment I and II ($R^2$ = 0.5) (*Figure 4b*). We labeled all of the interlinks that exhibit more than a twofold abundance difference in both experiments (*Figure 4b*). The crosslinks between Rpb4:80-Rpb7:23, Rpb4:76-Rpb7:23 and Rpb1:2-Rpb7:29, labeled in green, were more abundant in the holo-pol II sample as the ΔRpb4 pol II sample lacks Rpb4 and contains sub-stoichiometric amounts of Rbp7. Several interlinks and intralinks involving Rpb1:332, Rpb2:507, and Rpb1:1102, labeled in red, are also more abundant in holo-pol II (*Figure 4b and c*). Rpb1:332 is located in the Switch 2 region (Rpb1:328–346), which is the main switch responsible for the conformational change from the 'open' to 'closed' state (*Figure 5a*). The helix in the open state (cyan) flips out (gray) toward the cleft to contact DNA at the −2, −1, and +2 positions (*Gnatt et al., 2001*). This

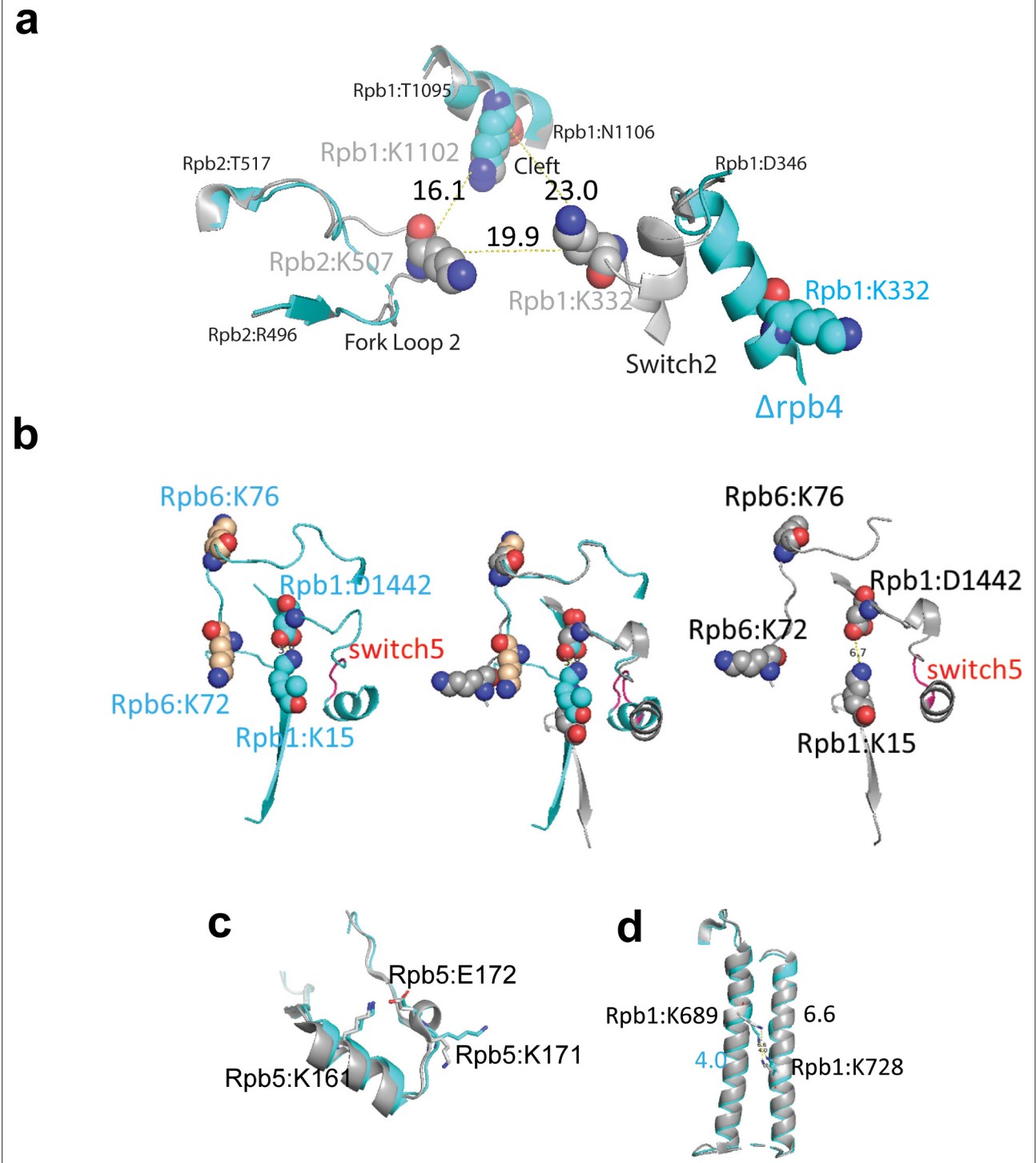

**Figure 5.** Q2linkers detect conformational changes in pol II complexes. (**a**) Switch 2 (Rpb1:328–346) conformational changes in ΔRpb4-pol II (cyan) and holo-pol II (gray). The crosslinked lysine residues are shown as spheres. Cα−Cα distances between crosslinked lysines are indicated. (**b**) Structural comparison of crosslinks involving Switch 5 (red, Rpb1:1431–1433) for ΔRpb4-pol II (cyan, left), holo-pol II (gray, right), and the merged structures (middle). In the holo-pol II structure (right), Switch 5 bending pulls Rpb1:D1442 away from K15, breaking the salt bridge that is formed in the core pol II structure (left). The increase in the abundances of the Rpb1:15-Rpb6:76 and Rpb1:15-Rpb6:72 crosslinks in holo-pol II is likely attributed to the salt bridge between K15 and D1442 in core pol II, which impedes the NHS ester-based reaction between the epsilon amino group of K15 and the crosslinker. (**c**) Structural comparison of the region of Rpb5 involving the crosslink between K161 and K171 in ΔRpb4-pol II and holo-pol II. A salt bridge (not shown) is formed between K161 and E172 in both structures. (**d**) Structure of Rpb1 near the crosslink between Rpb1:K689 and Rpb1:K728 in ΔRpb4-pol II and holo-pol II. In all structures, ΔRpb4-pol II is cyan and holo-pol II is gray. NZ−NZ distances between crosslinked lysines are indicated.

conformational change is accompanied by stabilization of the Rpb2 forkloop 2 (Rpb2:503–508). The conformational change in Switch 2 brings Rpb1:332, Rpb1:1102, and Rpb2:507 closer to each other, allowing them to be crosslinked by Q2linkers (*Figure 5a*).

The crosslinks between Rpb1:15-Rpb6:76 and Rpb1:15-Rpb6:72, labeled in blue, are influenced by a conformational change in Switch 5 (Rpb1:1431–1433), which undergoes a hinge-link bend (*Gnatt et al., 2001*; *Figure 5b*). In the core pol II structure, Rpb1:K15 forms a salt bridge with Rpb1:D1442 (K15:NZ - D1442:OD2 distance of 3.8 Å). In holo-pol II, Switch 5 bending pulls Rpb1:D1442 away from K15, breaking the salt bridge (NZ-OD2 distance of 6.7 Å). Even though the relative positions of Rpb1:K15, Rpb6:K76, and Rpb6:K72 do not change significantly in the holo- and core pol II structures (*Figure 5b*), the increase in crosslink abundances involving these residues in the 'open' state is likely attributed to the salt bridge between K15 and D1442 in the 'closed' state, which impedes the NHS ester-based reaction between the epsilon amino group of K15 and the crosslinker. Not all of the crosslink abundance changes can be easily explained. Rpb1:1246 is located in a region that interacts with Rpb9 and Rpb2, and changes from a disordered region (1245–1254) in core pol II to a loop in holo-pol II (*Gnatt et al., 2001*). The increased abundance of the Rpb1:1246-Rpb9:77 in holo-pol II might reflect this structural change. Rpb10:68 is very close to Rpb2:191, but there is no obvious difference in the location of the residues in both structures. Rpb1:129 is located at the N-terminus of Rpb1 (the major part of the clamp domain 1–346) that moves as a rigid body. A slight orientation difference of Rpb1:K129 may affect its ability to crosslink to Rpb5:K171. The crosslink between Rpb11:37-Rpb1:129 could be a false-positive identification as these residues are located on opposite sides of pol II and the distance between their Cα atoms exceeds the theoretical crosslinking distance of Q2linkers.

Here, 287 intralinks were identified in both experiments, and 124 and 143 intralinks were identified only in experiments I and II, respectively (*Figure 4c*). The anti-correlation of log2(126/127) ratios for the intralinks identified in both experiments is not strong ($R^2 = 0.3$). All of the intralinks involving Rpb4 are more abundant in holo-pol II sample (even though we do not expect any reporter ion signal from Rpb4 peptides derived from the ΔRpb4 pol II sample, we still observed reporter ion signals from the channel corresponding to the ΔRpb4 sample, potentially due to the presence of low abundance, co-eluting ions). In this study, the intralinks are less useful for probing conformational changes as the backbones of most subunits align pretty well in both complexes, and there are only small rearrangements for most residues. For example, the Rpb5:161–171 crosslink is enriched in the ΔRpb4-pol II sample, but there is little structural change for the region involved (*Figure 5c*). Rpb5:K161 overlays perfectly in both structures and forms a salt bridge with Rpb5:E172 with NZ-OE2 distances of 2.7 Å and 2.8 Å in the holo- and core pol II structures, respectively (*Figure 5c*). Rpb5:K171 assumes slightly different orientations in both structures due to the formation of a helical structure between residues 171–175 in holo-pol II (*Figure 5c*). Rpb5:K171 is slightly more stable in core pol II (cyan, b-factor 74) than in holo-pol II (gray, b-factor 216), which may account for its increased abundance in the ΔRpb4-pol II sample. The Rpb1:689–728 crosslink is enriched in the holo-pol II sample, while these two sites, located in the middle of two alpha-helices, are well-aligned (*Figure 5d*). The distance between the two NZ atoms of the lysine residues is ~4 Å in core-pol II and ~6.6 Å in holo-pol II. The 4 Å distance in the ΔRpb4-pol II sample may limit the ability of the Q2linkers to react with both ε amines to form a crosslink between the two lysine side chains. Also, 191 monolinked peptides were identified in both experiments and 10 and 11 monolinked peptides were only identified in experiment I or II, respectively. All of the Rpb4 and Rpb7 monolinks are enriched in the holo-pol II sample as expected. Some Rpb1 residues at the N-terminal clamp domain were also enriched in the holo-pol II sample (*Figure 4d*). In general, and like intralinks, the monolinks are less informative in this study for probing conformational changes. Interestingly, some Rpb5 monolinks and intralinks are slightly enriched in the ΔRpb4-pol II sample, even though there is no large conformational or structural rearrangement associated with this region. Rpb5 interacts with the same domains of Rpb6 and Rpb1 that interact with Rpb4 and Rpb7. It is possible that Rpb4/7 stabilizes Rpb1, Rpb6, and Rpb5 and their interactions with one another so that the region is more rigid in holo-pol II and less reactive to Q2linkers.

## Discussion

In this article, we describe a new approach for studying conformational and structural changes in proteins and protein complexes that is based on quantitative CLMS with a novel set of isobaric crosslinking reagents, called Q2linkers. Q2linkers are small and simple, amine-reactive molecules with an

optimal extended distance of ~10 Å for CLMS. After crosslinking, the samples are combined for all subsequent steps in the analysis, including enzymatic digestion, peptide fractionation/clean-up, and MS analysis, thus minimizing variations that may occur during these steps. The ability to avoid technical biases introduced during sample processing is especially important in qCLMS studies where conformational changes may be revealed by small but reproducible quantitative changes in crosslinking efficiency. The MS2-based reporter ion quantification is simple and compatible with most high-resolution mass spectrometers. The isobaric qCLMS technology can capture both conformational changes and structural re-arrangements in complex protein samples, and is well-suited to be adopted as a common strategy to study protein–protein interactions and conformational changes in large protein complexes under different conditions.

Isobaric crosslinkers (*Chavez et al., 2020*; *Chavez et al., 2021*) are attractive for studying conformational and structural changes for a number of reasons. Unlike label-free approaches, they allow samples to be combined immediately after crosslinking and analyzed by MS together, thus avoiding potential quantification inaccuracies due to artifacts that can occur during sample processing such as different digestion efficiencies, differential sample losses during additional isotopic labeling and/ or purification steps, or different extents of amino acid modifications (i.e., methionine oxidation, or N-terminal glutamate to pyroglutamate conversion). In addition, quantification of crosslinked peptides based on isotope labeling has been shown to be more accurate than label-free based methods (*Walzthoeni et al., 2015*), which likely is important for detecting subtle conformational changes. Finally, unlike approaches that employ isotopically heavy and light isotopes, isobaric crosslinkers do not increase the complexity of the MS1 spectra, which may improve sensitivity and reproducibly. Our Qlinker design is one of the simplest for isobaric crosslinkers. While the basic structure could be expanded to iminodiacetic acid and iminodibutyric acid for specific applications, we think the iminopropionic-based Qlinker design may be the most useful for general practice.

One of the biggest challenges associated with CLMS technology is how to interpret and use the resulting distance restraints. The crosslinking results are often used in integrative modeling approaches to produce low- to medium-resolution structural models or for verification of cryo-EM structures (*Chavez et al., 2015*; *Abdella et al., 2021*; *Patel et al., 2019*). It is even more challenging to use the crosslinking results to study conformational and structural changes. As we have shown in the ΔRpb4-pol II and holo-pol II experiments, many interlinks (32–53%) and intralinks (30–33%) were only identified in one of the experiments (*Figure 4*). Most of the crosslinks identified in only one experiment showed no significant abundance change in the two samples, suggesting that their identification in only one experiment is likely due to undersampling during MS analysis of the complex samples rather than the presence of the crosslinked peptides in only one experiment. This, combined with the potential technical issues associated with label-free qCLMS analyses mentioned above, highlights some of the challenges associated with inferring conformational changes based on label-free qCLMS data. Previously, we performed triplicate label-free qCLMS experiments to study structural changes in the histone octamer upon ISW2 interaction (*Hada et al., 2019*). To alleviate issues due to undersampling, we only considered crosslinked peptides identified in at least two experiments for inferring conformational changes. Unfortunately, this strategy becomes less effective as sample complexity increases and undersampling issues are exacerbated. While computational approaches that align MS runs and match MS1 features across MS runs provide a way to alleviate undersampling issues (*Walzthoeni et al., 2015*; *Valot et al., 2011*; *MacLean et al., 2010*), these approaches are not ideal. The Q2linker, isobaric crosslinker-based qCLMS strategy alleviates issues due to undersampling and sample handling that are often encountered with label-free approaches, and thus permits reliable identification and quantification of site-specific changes in crosslinker reactivity associated with structural differences in two samples.

In large protein complexes, such as the pol II complex, interlinks are more informative than intralinks and monolinks for inferring structural changes since they provide information about the proximities of the modified amino acids and their associated domains. MS-based technologies such as H/D exchange and active hydroxyl radical mapping can provide information about changes in surface exposure and residue accessibilities (*McKenzie-Coe et al., 2022*; *Zheng et al., 2019*; *Tsirigotaki et al., 2017*; *Huang et al., 2015*). However, in large protein complexes, changes in surface exposure and residue accessibility are often not associated with structural shifts. For example, compared to ΔRpb4-pol II, there is a large 'clamp' domain movement in holo-pol II, but the clamp moves as a

rigid body and this large structural rearrangement is controlled by small conformational changes of several 'switches'. The monolinks are not informative as the accessibility of surface exposed lysine residues does not significantly change. In fact, even though there is a >30 Å movement at the tip of the clamp, we did not detect significant ratio changes for the crosslinks that map to the clamp domain because the conformation of the domain itself does not change significantly. Instead, we observed changes in the abundances of crosslinks involving residues associated with the switches that cause the large-scale movement. Isobaric qCLMS technology provides a powerful way to characterize conformational changes in protein complexes that is difficult to achieve using currently available structural-MS- approaches. In addition, the Q2linker approach complements the information provided by high-resolution structural approaches such as cryo-EM. Unlike cryo-EM or X-ray crystallography, qCLMS is performed in solution under near physiological conditions. Furthermore, CLMS can provide structural information about flexible or disordered regions, which is often difficult to obtain by high-resolution approaches. However, as the success of CLMS very much depends on the experimental conditions and we cannot predict which crosslinked peptides can be identified, it is very difficult to predict what kinds of changes qCLMS can detect. Additional applications of Qlinker-CLMS are needed to better understand the types of structural changes that can be detected using the approach.

We found that one of the most important considerations in the design of quantitative CLMS experiments, unexpectedly, is to have a situation where most of the ratios are close to 1:1. If the ratios are skewed toward one condition, it becomes very difficult to interpret the ratio changes. Many steps during sample preparation can result in sample-specific abundance changes, which are unrelated to conformational or structural differences. This can occur even after mixing the crosslinked samples and processing them together. For example, we observed that small proteins were adversely lost during the SP3 protein enrichment step, resulting in high abundance ratios for all crosslinker-modified peptides derived from these small proteins in samples where they interact to form larger complexes. To alleviate this issue, we tried many other methods to denature and enrich proteins before trypsin digestion or direct trypsin digestion without enrichment, such as 1% SDS, TCA precipitation, 50% TFE, 8 M urea, and found that 1% SDC gave us the best results for both peptide recovery and crosslinked peptide identification. Thus, we recommend using 1% SDC to denature crosslinked samples prior to trypsin digestion, especially when the samples contain small proteins. For large protein complexes, the SP3 strategy is also preferable due to its simplicity.

Like most CLMS approaches, this isobaric qCLMS approach depends on the modification of specific chemical moieties and detection of the modified peptides. It may not capture conformational changes if there are no reactive groups near the region involved in the structural shift or the modified peptides are difficult to detect during MS analysis. We did not observe changes in crosslinker-modified peptide abundances derived from pol II in the presence and absence of α-amanitin. This implies that this strategy, like all MS-based strategies, can only be used for interpretation of positively identified crosslinks or monolinks. Sensitivity and undersampling are common problems for MS analysis of complex samples. Future development of Qlinker will involve the generation of affinity reagents that can enrich the crosslinker-modified peptides to improve sensitivity and quantification, and incorporation of multiple isobaric labels in the Qlinkers, like the TMT labels, so that multiple conditions and replicates can be analyzed simultaneously.

## Materials and methods

### Key resources table

| Reagent type (species) or resource | Designation | Source or reference | Identifiers | Additional information |
|---|---|---|---|---|
| Strain, strain background (*Escherichia coli*) | BL21(DE3)-CodonPlus-RIL | Agilent | Cat# 230245 | Chemically Competent cells |
| Strain, strain background (*Saccharomyces cerevisiae*) | BY4741 with C-terminal His6-3XFLAG-His6-Ura3 (HFH) tag on RPA2 | This paper | | Available on request from the Ranish lab |
| Strain, strain background (*S. cerevisiae*) | BY4741 with C-terminal (HFH) tag on RPB3 | This paper | | Available on request from the Ranish lab |
| Strain, strain background (*S. cerevisiae*) | Δ*rpb4* strain with C-terminal (HFH) tag on RPB2 | This paper | | Available on request from the Ranish lab |

*Continued on next page*

*Continued*

| Reagent type (species) or resource | Designation | Source or reference | Identifiers | Additional information |
|---|---|---|---|---|
| Recombinant DNA reagent | His6-tagged TBP | Dr. Steven Hahn (Fred Hutchinson Cancer Research Center) | | |
| Recombinant DNA reagent | His6-tagged TFIIB | Dr. Steven Hahn (Fred Hutchinson Cancer Research Center) | | |
| Recombinant DNA reagent | His6-tagged TFIIA | PMID:28259734 | | |
| Peptide, recombinant protein | Human Brain Calmodulin | MilliporeSigma | Cat# 208698500ug | |
| Peptide, recombinant protein | Calmodulin binding peptide 1 | GenScript Biotech Corp. | Cat# RP13247 | |
| Peptide, recombinant protein | Bacterial maltose binding protein | Novus Biologicals | Cat# NBC118538 | |
| Chemical compound, drug | Di-tert-butyl 3,3'-Iminodipropionate | TCI America | D4110 | CAS: 128988-04-5 |
| Chemical compound, drug | Bromoacetic acid (1-$^{13}$C) | Cambridge Isotope Laboratories, Inc | CLM-723-PK | CAS: 57858-24-9 |
| Chemical compound, drug | Bromoacetic acid (2-$^{13}$C) | Cambridge Isotope Laboratories, Inc | CLM-724-PK | CAS: 64891-77-6 |
| Chemical compound, drug | *cis*-2'–6'-Dimethylpiperidine | MilliporeSigma | D180300 | CAS: 766-17-6 |
| Software, algorithm | Nexus | PMID:30343899 | | Source code 1 |
| Software, algorithm | pLink2 | PMID:31363125 | | |
| Software, algorithm | Rawconverter | PMID:26499134 | | |
| Software, algorithm | Trans-Proteomics Pipeline (TPP)/ | PMID:36648445 | | http://tools.proteomecenter.org/wiki/index.php?title=Software:TPP |

## Materials

Di-tert-butyl 3,3'-iminodipropionate was purchased from TCI America (Portland, OR). Bromoacetic acid 1-$^{13}$C and 2-$^{13}$C were purchased from Cambridge Isotope Laboratories, Inc (Tewksbury, MA). The N,N'–diisopropylcarbodiimide (DIC), N,N-diisopropylethylamine (DIPEA), dimethylformamide (DMF), acetonitrile (ACN), dichloromethane (DCM) 2,6-dimethylpiperidine, N,N,N',N'-Tetramethyl-O-(N-succinimidyl)uronium tetrafluoroborate (TSTU), Human Brain Calmodulin, and bovine serum albumin (BSA) were purchased from MilliporeSigma (St. Louis, MO). Calmodulin binding peptide 1 (alphaII-spectrin peptide) was purchased from GenScript Biotech Corp. (Piscataway, NJ). Bacterial maltose binding protein was purchased from Novus Biologicals (Littleton, CO). NGC chromatography system from Bio-Rad Laboratories (Hercules, CA) was used for reverse-phase FPLC-C18 separation, and SNAP ULTRA C18 flash cartridges (40 g) were purchased from Biotage (Uppsala, Sweden).

## Synthesis of *bis*(succinimidyl)-3,3'-{[(2,6-dimethylpiperidin-1-yl)acetyl]azanediyl}dipropanoic acid ('Q2linker')

To synthesize the Qlinker, we first synthesized (2,6-dimethylpiperidin-1-yl) acetic acid and reacted it with di-tert-butyl-3,3'-iminodipropoinate ('**1**') but we obtained little desired product due to steric hindrance of dimethylpiperidine, resulting in difficulties activating the (2,6-dimethylpiperidin-1-yl) acetic acid. Then we used a peptoid synthesis strategy to have a one-pot synthesis of di-tert-butyl protected product '**2**' using the 1-$^{13}$C or 2-$^{13}$C bromoacetic acid (BAA) for the two isobaric cross-linkers (*Figure 1—figure supplement 1*). 0.5 mL di-tert-butyl 3,3'-iminodipropionate ('**1**') (~3.2 M, 1 eq), 0.5 ml DIC (~6.4 M, 2 eq), and 280 mg bromoacetic acid 1-$^{13}$C or 2-$^{13}$C (1.25 eq) in 1 ml DMF were mixed for 2 hr at room temperature (RT). Then 10 ml DCM and 10 ml 0.1% trifluoroacetic acid (TFA) were used to extract product di-tert-butyl-3,3'-[(bromoacetyl)azanediyl]dipropanoic acid in the DCM phase. 1.1 ml (~5 eq) dimethylpiperidine was added to the DCM extract and mixed at RT for 2 hr, and the solvent was then evaporated under vacuum. The product di-tert-butyl-3,3'-{[(2,6-dimethylpiperidin-1-yl)acetyl]azanediyl}dipropanoic acid '**2**' was then dissolved in 50% ACN/0.1% TFA and diluted to 20% ACN. The precipitates were removed by centrifugation and the supernatant was loaded onto a FPLC-RP-C18 cartridge at 5 ml/min and eluted with a 70 min gradient from 20 to 60% ACN. The product was monitored by 215 nm absorption and confirmed by MS analysis using an LTQ (Thermo Scientific). The fractions containing the product '**2**' were then combined and evaporated by

rotovap. 5 ml TFA was then added to the dried substance for 2 hr and then evaporated by rotovap. The final product '**3**' of 3,3'-{[(2,6-dimethylpiperidin-1-yl)acetyl]azanediyl}dipropanoic acid was then dissolved in 0.1% TFA, loaded onto an FPLC-RP-C18 column, and eluted with a 50 min gradient from 0 to 20% ACN. Product '**3**' with MH+ of 316.34 was collected and was determined to be >95% pure. Product '**3**' was fractionated a second time using the same FPLC-RP-C18 conditions to yield ~50 mg of the final product at ~99% purity (~10% efficiency). In situ activation of Q2linker was achieved by mixing 0.25 M product '**3**' dissolved in dry DMF with an equal volume of 0.5 M TSTU in dry DMF and 1/10th volume of DIPEA at RT for 1 hr. The final stock concentration of Q2linkers was ~0.12 M, and the crosslinkers can be stored at –20°C or –80°C for up to 1 month with little loss of activity. Hydrolysis of the activated crosslinker was observed upon longer storage, possibly due to trace amounts of water in the DMF solution.

## General crosslinking protocol and workflow optimization

To perform the crosslinking reaction, we typically used freshly activated crosslinkers or brought the frozen crosslinkers to RT for 30 min before use. A typical crosslinking reaction is carried out in 50 mM HEPES buffer (pH 7.9) or 1× PBS (pH 7.5) with 20–100 µg total protein in a volume of 50–400 µl with ~2 mM (50× dilution from the stock) crosslinker. Nearly equal amounts of proteins were first crosslinked with one of the Q2linkers for 1–2 hr at RT and then 10 µl 1 M ammonium sulfate was added to quench each reaction for 10 min. The two crosslinked samples were then combined and vortexed to mix the sample. 10 µl 20 mg/ml SP3 magnetic beads were added to the sample and the sample was mixed well. ACN was then added to 70%, and the sample was incubated in a thermomixer at 60°C for 30 min with mixing. After collecting the beads on a magnetic stand, the beads were washed with 100% ACN and resuspended in 100 µl 8 M urea, 50 mM choloroacetamide (CAA), 50 mM TCEP in 1 M ammonium bicarbonate buffer at 37°C for 1 hr. The beads were then diluted by adding 700 µl pure water and 1/10 (w/w) trypsin to digest the sample overnight at 37°C with rotation. The digested peptides were then purified using C18 micro-tips and dried for MS analysis. Complex samples were fractionated by HPLC using in-house prepared microcapillary SCX columns (200 µm × 20 cm; SCX 3 µm, Sepax Technologies) at a flow rate of 2–3 µl/min. Peptides were eluted with 20 µl of Buffer A (10% ACN, 0.1% FA) containing 30, 50, 70, and 100% Buffer B (800 mM ammonium formate, 20% ACN, pH 2.8), followed by 50 µl Buffer D (0.5 M ammonium acetate, 30% ACN). All fractions were dried in a Speed-vac and resuspended in 0.1% TFA and 2% ACN.

We used the C1q2 and C2q2 reagents to crosslink BSA and analyzed the sample using a Thermo Oribtrap-Fusion mass spectrometer with an HCD collision energy of 28 or 30%. The monolinked peptide spectra were readily identified by Comet database searching using a differential modification of 297.1770 on lysine residues, but most spectra had very low or no reporter ion intensities. We then increased the HCD collision energy to 35, 40, and 45% and observed higher reporter ion intensities, with 35% energy giving both reasonable fragmentation patterns and high reporter ion intensities, and collision energies above 35% resulting in poor fragmentation spectra and significantly reduced the numbers of identified peptides (data not shown). Synchronous precursor selection coupled with MS3 (SPS-MS3) technology can be used to quantify TMT-labeled peptides because all of the b ions and lysine containing y ions are TMT-labeled and can be selected for MS3 to generate reporter ions at high energy (*McAlister et al., 2014*). However, this technology cannot be used for Q2linker quantification because most fragment ions will not retain the reporter moiety and thus will not yield reporter ions during MS3. At the same time, the Q2linkers are only associated with modified lysine residues and may not produce observable daughter ions in the MS2 spectra that can be selected for MS3, especially for the crosslinked species. The Yates group reported that a stepped HCD strategy increased both the diversity of fragmentation ions and TMT reporter ion intensity (*Diedrich et al., 2013*). Following their suggestion for TMT-labeled peptides, we used stepped HCD settings of 24, 30, and 36% and obtained MS2 spectra with much higher reporter ion intensities as well as better fragmentation across the peptide backbone. The average reporter ion intensity under this condition is about 60% ± 25% of the highest intensity peaks. Thus, this condition was used for all subsequent MS analyses of Q2linker-crosslinked samples.

## Purification of TFIIA, TFIIB, and TBP proteins

Expression plasmids containing His6-tagged versions of TBP, TFIIA, and TFIIB were transformed into BL21 (DE3)-CodonPlus-RIL cells (Agilent, Santa Clara, CA) in LB broth with carbenicillin/kanamycin and chloramphenicol. Protein expression was induced by addition of IPTG to 0.15 mM for TFIIA/TFIIB and 0.5 mM for TBP at 37°C for 4 hr after OD600 reached 0.3–0.5. Cells were lysed in lysis buffer (20 mM Tris-HCl pH 7.5, 250 mM KCl, 10% glycerol, 10 mM βME, 10 mg/ml lysozyme) for 30 min at 4°C and then sonicated using a Superhorn sonicator with output 9 in an ice-water bath using 1 min cycles of 30 son, 30 s off, over 10 min. The lysates were clarified by centrifugation at 30,000 × $g$ for 20 min and proteins were purified using HisPur Ni-NTA resin (Thermo Fisher Scientific). The proteins were eluted with 200 mM imidazole and concentrated using 3K MWCO Amicon Ultra 4 ml spin columns followed by buffer-exchange to reaction buffer (20 mM HEPES-NaOH, pH 7.9, 100 mM KOAc). 25U of SUMO protease (Invitrogen) was used to cleave the N-terminal His-sumo tag on TFIIB at 4°C overnight and HisPur Ni-NTA was used to remove the SUMO protease and the His6-sumo tag.

## Qlinker analysis of complexes containing TBP, TFIIA, and TFIIB

20 ug of each purified protein was incubated in 100 ul reaction buffer for 30 min at RT in the following combinations: TBP/TFIIA, TBP/TFIIB, and TBP/TFIIA/TFIIB. Then C1q2 crosslinkers were added to reactions containing TBP/TFIIA and TBP/TFIIB and C2q2 crosslinkers were added to reactions containing TBP/TFIIA/TFIIB at ~1 mM for 2 hr before quenching with 5 ul 1 M ammonium bicarbonate. The samples were combined and 1/10th volume of 10% sodium deoxycholate (SDC) was added followed by addition of TCEP and CAA to 10 and 25 mM, respectively. The proteins were denatured at 95°C for 10 min and then diluted to 0.5% SDC for trypsin digestion overnight. The SDC was precipitated by addition of 1% TFA and centrifugation at 16,000 × $g$ for 10 min. The peptides were C18 cleaned and fractionated by HPLC-SCX prior to MS analysis.

## Purification of yeast pol I and pol II complexes

We constructed yeast (*Saccharomyces cerevisiae*) strains carrying C-terminal His6-3XFLAG-His6-Ura3 (HFH) tags on *RPA2* and *RPB3* by swapping the tandem affinity purification (TAP) tag for the HFH tag in strains carrying C-terminal TAP tags on *RPA2* and *RPB3*. The *RPB2-HFH* strain was generated by transforming a *Δrpb4* strain with a PCR product containing the HFH tag with 40 bps of sequence flanking the stop codon of *RPB2* gene. For each complex purification, we normally grew 6 l of the appropriate yeast strains in YPD media overnight to OD600 11–13. The cells were harvested by centrifugation and frozen in liquid $N_2$. After evaporation of the liquid $N_2$, the cell pellets were ground to a fine powder in a coffee grinder. 40 ml lysis buffer (50 mM HEPES, pH 7.9, 400 mM ammonium sulfate, 10 mM $MgSO_4$, 1 mM EDTA, 20% glycerol) with protease inhibitors was then added to the fine powder and the mixture was stirred at 4°C for 1 hr. The cell lysate was then sonicated using a Superhorn sonicator with output 9 in an ice-water bath using 1 min cycles of 30 s on, 30 s off, over 10 min. The lysate was centrifuged at 20,000 × $g$ for 1 hr and the supernatant was then mixed with 2 ml anti-FLAG M2 affinity agarose (Sigma-Aldrich) overnight at 4°C with rotation. The beads were then collected in a column and washed with 40 ml lysis buffer twice, 40 ml 2× PBS buffer, and 40 ml 1× PBS with 0.1% NP-40. The complexes were eluted with 3X FLAG peptide (Sigma-Aldrich) at 0.4 mg/ml in 1× PBS. The complexes were then concentrated by repeated centrifugation and dilution with 1× PBS (normally three times) in Amicon Ultra-4 devices (100K cutoff, Millipore) to reduce the concentration of the 3X FLAG peptide. After a final centrifugation step, protein concentration was determined by Qubit protein assay (Thermo Fisher Scientific). We usually isolated ~200–300 µg protein from a 6 l culture. Higher yields of pol II were obtained from the *ΔRPB4* strain. We checked the purity of the sample and subunit composition by Coomassie stained SDS-PAGE.

## MS analysis and data processing

Peptides were analyzed by electrospray ionization microcapillary reverse-phase HPLC with a column (75 µm × 270 mm) packed with ReproSil-Pur C18AQ (3 µm 120 Å resin; Dr. Maisch, Baden-Würtemburg, Germany) on a Thermo Scientific Fusion with HCD fragmentation and serial MS events that included one FTMS1 event at 30,000 resolution followed by FTMS2 events at 15,000 resolution. Other instrument settings included MS1 scan range (m/z): 400–1500; cycle time 3 s; charge states 3–8; filters MIPS on, relax restriction = true; dynamic exclusion enabled: repeat count 1, exclusion duration 30 s; filter

IntensityThreshold, signal intensity 50,000; isolation mode, quadrupole; isolation window 3 Da; activation type: HCD; collision energy mode: stepped; HCD collision energy (%): 24, 30, 36; AGC target 500,000, max injection time 200 ms. HPLC uses an 80 min gradient from 10% ACN to 40% ACN.

The RAW files were converted to mzXML files by Rawconverter (*He et al., 2015*). For normal peptide and monolinked peptide searches, we used the Trans-Proteomics Pipeline (TPP)/Comet searches (http://tools.proteomecenter.org/wiki/index.php?title=Software:TPP) with static modification on cysteines (+57.0215 Da) and differential modifications on methionines (+15.9949 Da) and lysines (+297.166962 Da and +279.166397 Da). Spectra identified as peptides with lysine modification(s) with PeptideProphet probability >95% were output as spectra for monolinked peptides. For crosslinked peptide searches, we used two crosslink database searching algorithms: pLink2 (*Chen et al., 2019*) and an in-house-designed Nexus (*Mashtalir et al., 2018*) with Q2linker mass of 279.1664 Da against a database containing only yeast pol I or pol II protein sequences and its reverse decoy database(s). Other searching parameters include precursor monoisotopic mass tolerance: ±20 ppm; fragment mass tolerance: ±20 ppm; up to three miscleavages; static modification on cysteines (+57.0215 Da); differential oxidation modification on methionines (+15. 9949 Da), peptide N-terminal glutamic acid (–18.0106 Da), or N-terminal glutamine (–17.0265 Da); and Q2linker modification on lysines (+297.166962 Da). After performing the pLink2 and the Nexus analyses with 5% FDR, the search results were combined and each spectrum was manually evaluated for the quality of the match to each peptide using the COMET/Lorikeet Spectrum Viewer (TPP). The crosslinked peptides are considered confidently identified if at least four consecutive b or y ions for each peptide are observed and the majority of the observed ions are accounted for. Search results that did not meet these criteria were removed. The spectra that passed our evaluation are summarized in Table S1 and are uploaded into ProXL (*Riffle et al., 2016*) for viewing and data analysis. All of the data, including the spectra, linkages, and structural analyses, can be visualized at https://www.yeastrc.org/proxl_public/viewProject. do?project_id=634. The raw files are deposited at protomeXchange: PXD035939 and PXD056825.

## Q2linker quantification

A Perl script is used to extract the 126 and 127 reporter ion intensities for the identified monolinked- or crosslinked-spectra from mzXML files within a mass tolerance of 0.005 Da (~40 ppm) of the theoretical 1+ mass of 126.127726 and 127.131081 for the 126 and 127 reporter ions, respectively. These observed intensities are designated **I**126 and **I**127, respectively. Like TMT quantification, these observed intensities need to be adjusted based on the natural distribution of monoisotopic elements and the 99% purity of the heavy 13C element. It is relatively simple for the Q2linkers since only two channels need to be considered. We corrected the reporter ion intensities (**A**126 and **A**127) by solving the equations: **A**126 * 0.90754+**A**127 * 0.0 1 = **I**126 and **A**126 * 0.09246 + **A**127 * 0.90724 = **I**127. The final intensities **A**126 = (**I**126 - **I**127*0.01102)/0.906521 and **A**127 = (**I**127 - **I**126 * 0.10188)/0.906221 were used to calculate the log2 ratios of the126 and 127 reporter ions. If multiple spectra were identified for a crosslinker-modified site, we used the average log2 ratio of all the identified spectra corresponding to that site.

## Statistics and reproducibility

Average and boxplot in R were used to generate *Figure 1d and e*. Crosslinker swapping experiments were performed for the experiments presented in *Figures 2a and 4*.

## Acknowledgements

We thank Dr. Steve Hahn for the TBP and TFIIB expression plasmids, Dr. Toshiya Senda for the TFIIA expression plasmid, and Dr. Phil Gafken and Lisa Jones at the Fred Hutchinson Cancer Research Center Proteomics Core for assistance with mass spectrometry analysis. Funding support was provided by the NIH: RO1GM136974 to JR, RO1GM108908, and R01GM144559.

## Additional information

### Funding

| Funder | Grant reference number | Author |
|---|---|---|
| National Institutes of Health | RO1GM136974 | Jeff Ranish |
| National Institutes of Health | RO1GM108908 | Jeff Ranish |
| National Institutes of Health | R01GM144559 | Jeff Ranish |

The funders had no role in study design, data collection and interpretation, or the decision to submit the work for publication.

### Author contributions

Jie Luo, Conceptualization, Resources, Data curation, Software, Formal analysis, Funding acquisition, Investigation, Methodology, Writing - original draft, Writing - review and editing; Jeff Ranish, Supervision, Funding acquisition, Project administration, Writing - review and editing

### Author ORCIDs

Jie Luo http://orcid.org/0000-0003-2815-2682
Jeff Ranish https://orcid.org/0000-0001-7181-0287

Reviewer #1 (Public review): https://doi.org/10.7554/eLife.99809.3.sa1
Reviewer #2 (Public review): https://doi.org/10.7554/eLife.99809.3.sa2
Author response https://doi.org/10.7554/eLife.99809.3.sa3

## Additional files

### Supplementary files

• Supplementary file 1. Table containing information about the interlinks, intralinks, and monolinks reported in *Figure 4* . The information includes the peptide sequences and protein names, sites of Qlinker modification, score, search engine (N = Nexus, *P* = pLink2), reporter ion intensities, and reporter ion ratios.
• MDAR checklist
• Source code 1. Nexus algorithm.

### Data availability

The raw MS files for the data presented in figures 1, 3, 4 and 5 have been deposited to the ProteomeXchange Consortium via the PRIDE partner repository with the dataset identifiers PXD056825 (Figures 1 and 3) and PXD035939 (Figures 4 and 5). The datasets generated during and/or analyzed during the current study are included in the manuscript and supporting files. The Nexus algorithm is available in *Source code 1*.

The following datasets were generated:

| Author(s) | Year | Dataset title | Dataset URL | Database and Identifier |
|---|---|---|---|---|
| Luo J, Ranish J | 2024 | Isobaric crosslinking mass spectrometry technology for studying conformational and structural changes in proteins and complexes | https://www.ebi.ac.uk/pride/archive/projects/PXD056825 | PRIDE, PXD056825 |

*Continued on next page*

*Continued*

| Author(s) | Year | Dataset title | Dataset URL | Database and Identifier |
|-----------|------|---------------|-------------|-------------------------|
| Luo J, Ranish J | 2024 | An isobaric quantitative crosslinking mass spectrometry technology for studying conformational and structural changes in proteins and protein complexes | https://www.ebi.ac.uk/pride/archive/projects/PXD035939 | PRIDE, PXD035939 |

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
