## [Editor Report · eLife Assessment]

This article presents a **valuable** new quantitative crosslinking mass spectrometry approach using novel isobaric crosslinkers. The data are **solid** and the method has potential for a broad application in structural biology if more isobaric crosslinking channels are available and the quantitative information of the approach is exploited in more depth.

---

## [Referee Report · Reviewer #1 (Public review)]

Summary:

Crosslinking mass spectrometry has become an important tool in structural biology, providing information about protein complex architecture, binding sites and interfaces, and conformational changes. One key challenge of this approach represents the quantitation of crosslinking data to interrogate differential binding states and distributions of conformational states.

Here, Luo and Ranish present a novel class of isobaric crosslinkers ("Qlinkers"), conduct proof-of-concept benchmarking experiments on known protein complexes, and show example applications on selected target proteins. The data are solid and this could well be an exciting, convincing new approach in the field if the quantitation strategy is made more comprehensive and the quantitative power of isobaric labeling is fully leveraged as outlined below. It's a promising proof-of-concept, and potentially of broad interest for structural biologists.

Strengths:

The authors demonstrate the synthesis, application, and quantitation of their "Q2linkers", enabling relative quantitation of two conditions against each other. In benchmarking experiments, the Q2linkers provide accurate quantitation in mixing experiments. Then the authors show applications of Q2linkers on MBP, Calmodulin, selected transcription factors, and polymerase II, investigating protein binding, complex assembly, and conformational dynamics of the respective target proteins. For known interactions, their findings are in line with previous studies, and they show some interesting data for TFIIA/TBP/TFIIB complex formation and conformational changes in pol II upon Rbp4/7 binding.

Weaknesses:

This is an elegant approach but the power of isobaric mass tags is not fully leveraged in the current manuscript.

First, "only" Q2linkers are used. This means only two conditions can be compared. Theoretically, higher-plexed Qlinkers should be accessible and would also be needed to make this a competitive method against other crosslinking quantitation strategies. As it is, two conditions can still be compared relatively easily using LFQ - or stable-isotope-labeling based approaches. A "Q5linker" would be a really useful crosslinker, which would open up comprehensive quantitative XLMS studies.

Second, the true power of isobaric labeling, accurate quantitation across multiple samples in a single run, is not fully exploited here. The authors only show differential trends for their interaction partners or different conformational states and do not make full quantitative use of their data or conduct statistical analyses. This should be investigated in more detail, e.g. examine Qlinker quantitation of MBP incubated with different concentrations of maltose or Calmodulin incubated with different concentrations of CBPs. Does Qlinker quantitation match ratios predicted using known binding constants or conformational state populations? Is it possible to extract ratios of protein populations in different conformations, assembly, or ligand-bound states?

With these two points addressed this approach could be an important and convincing tool for structural biologists.

Comments on latest version:

I raised only two points which they have not addressed: Higher multiplexing of Qlinkers (1) and experiments to assess the statistical power of their quantitation strategy (2).

I can see that point (1) requires substantial experimental efforts and synthesis of novel Qlinkers would be months of work. This is an editorial decision if the limited quantitative power of the "2-plex" approach they have right now is sufficient to support publication in eLife. While I like the approach, I feel it falls short of its potential in its current form.

For point (2), the authors did not do any supporting experiments. They claim "higher plex Qlinkers" would need to be available, but I suggested experiments that can be done even with Q2linkers: Using one of the two channels as a reference channel (similar the Super-SILAC strategy published in 2010 by Geiger et al; using an isotope-labeled channel as a stable reference channel between different experiments and LC-MS runs), they could do time-courses or ligand-concentration-series with the other channel and then show that Qlinkers allow quantitative monitoring of the different populations (e.g. conformations or ligand-bound proteins).

As an additional point, I was a bit surprised to read that the quantitation evaluation in Figure 1 is based on a single experiment (reviewer response document page 6, line 2 in the authors' reply). I strongly suggest this to be repeated a few times so a proper statistical test on experimental reproducibiltiy of Qlinkers can be conducted.

In summary, the authors declined to do any experimental work to address my concerns.

---

## [Referee Report · Reviewer #2 (Public review)]

The regulation of protein function heavily relies on the dynamic changes in the shape and structure of proteins and their complexes. These changes are widespread and crucial. However, examining such alterations presents significant challenges, particularly when dealing with large protein complexes in conditions that mimic the natural cellular environment. Therefore, much emphasis has been put on developing novel methods to study protein structure, interactions, and dynamics. Crosslinking mass spectrometry (CSMS) has established itself as such a prominent tool in recent years. However, doing this in a quantitative manner to compare structural changes between conditions has proven to be challenging due to several technical difficulties during sample preparation. Luo and Ranish introduce a novel set of isobaric labeling reagents, called Qlinkers, to allow for a more straightforward and reliable way to detect structural changes between conditions by quantitative CSMS (qCSMS).

The authors do an excellent job describing the design choices of the isobaric crosslinkers and how they have been optimized to allow for efficient intra- and inter-protein crosslinking to provide relevant structural information. Next, they do a series of experiments to provide compelling evidence that the Qlinker strategy is well suited to detect structural changes between conditions by qCSMS. First, they confirm the quantitative power of the novel-developed isobaric crosslinkers by a controlled mixing experiment. Then they show that they can indeed recover known structural changes in a set of purified proteins (complexes) - starting with single subunit proteins up to a very large 0.5 MDa multi-subunit protein complex - the polII complex.

The authors give a very measured and fair assessment of this novel isobaric crosslinker and its potential power to contribute to the study of protein structure changes. They show that indeed their novel strategy picks up expected structural changes, changes in surface exposure of certain protein domains, changes within a single protein subunit but also changes in protein-protein interactions. However, they also point out that not all expected dynamic changes are captured and that there is still considerable room for improvement (many not limited to this crosslinker specifically but many crosslinkers used for CSMS).

Taken together the study presents a novel set of isobaric crosslinkers that indeed open up the opportunity to provide better qCSMS data, which will enable researchers to study dynamic changes in the shape and structure of proteins and their complexes.

Comments on latest version:

The authors have not really addressed most of the concerns. They have added minimal discussion points to the text. This is okay from my perspective as eLife's policy is to leave it up to the authors of how strongly to consider the reviewers' comments. I should add that I do fully agree with the other reviewer that the quantitative assessment from Figure 1 should have been done in triplicates at least and that this would actually be essential.

---

## [Author Response]

The following is the authors’ response to the previous reviews.

**Reviewer #1 (Public review):**
Summary:Crosslinking mass spectrometry has become an important tool in structural biology, providing information about protein complex architecture, binding sites and interfaces, and conformational changes. One key challenge of this approach represents the quantitation of crosslinking data to interrogate differential binding states and distributions of conformational states.Here, Luo and Ranish present a novel class of isobaric crosslinkers ("Qlinkers"), conduct proof-of-concept benchmarking experiments on known protein complexes, and show example applications on selected target proteins. The data are solid and this could well be an exciting, convincing new approach in the field if the quantitation strategy is made more comprehensive and the quantitative power of isobaric labeling is fully leveraged as outlined below. It's a promising proof-of-concept, and potentially of broad interest for structural biologists.Strengths:The authors demonstrate the synthesis, application, and quantitation of their "Q2linkers", enabling relative quantitation of two conditions against each other. In benchmarking experiments, the Q2linkers provide accurate quantitation in mixing experiments. Then the authors show applications of Q2linkers on MBP, Calmodulin, selected transcription factors, and polymerase II, investigating protein binding, complex assembly, and conformational dynamics of the respective target proteins. For known interactions, their findings are in line with previous studies, and they show some interesting data for TFIIA/TBP/TFIIB complex formation and conformational changes in pol II upon Rbp4/7 binding.Weaknesses:This is an elegant approach but the power of isobaric mass tags is not fully leveraged in the current manuscript.First, "only" Q2linkers are used. This means only two conditions can be compared. Theoretically, higher-plexed Qlinkers should be accessible and would also be needed to make this a competitive method against other crosslinking quantitation strategies. As it is, two conditions can still be compared relatively easily using LFQ - or stable-isotope-labeling based approaches. A "Q5linker" would be a really useful crosslinker, which would open up comprehensive quantitative XLMS studies.

We agree that a multiplexed Qlinker approach would be very useful. The multiplexed Qlinkers are more difficult and more expensive to synthesize. We are currently working on different schemes for synthesizing multiplexed Qlinkers.

Second, the true power of isobaric labeling, accurate quantitation across multiple samples in a single run, is not fully exploited here. The authors only show differential trends for their interaction partners or different conformational states and do not make full quantitative use of their data or conduct statistical analyses. This should be investigated in more detail, e.g. examine Qlinker quantitation of MBP incubated with different concentrations of maltose or Calmodulin incubated with different concentrations of CBPs. Does Qlinker quantitation match ratios predicted using known binding constants or conformational state populations? Is it possible to extract ratios of protein populations in different conformations, assembly, or ligand-bound states?With these two points addressed this approach could be an important and convincing tool for structural biologists.

We agree that multiplexed Qlinkers would open the door to exciting avenues of investigation such as studying conformational state populations. We plan to conduct the suggested experiments when multiplexed Qlinkers are available.

**Reviewer #2 (Public review):**
The regulation of protein function heavily relies on the dynamic changes in the shape and structure of proteins and their complexes. These changes are widespread and crucial. However, examining such alterations presents significant challenges, particularly when dealing with large protein complexes in conditions that mimic the natural cellular environment. Therefore, much emphasis has been put on developing novel methods to study protein structure, interactions, and dynamics. Crosslinking mass spectrometry (CSMS) has established itself as such a prominent tool in recent years. However, doing this in a quantitative manner to compare structural changes between conditions has proven to be challenging due to several technical difficulties during sample preparation. Luo and Ranish introduce a novel set of isobaric labeling reagents, called Qlinkers, to allow for a more straightforward and reliable way to detect structural changes between conditions by quantitative CSMS (qCSMS).The authors do an excellent job describing the design choices of the isobaric crosslinkers and how they have been optimized to allow for efficient intra- and inter-protein crosslinking to provide relevant structural information. Next, they do a series of experiments to provide compelling evidence that the Qlinker strategy is well suited to detect structural changes between conditions by qCSMS. First, they confirm the quantitative power of the novel-developed isobaric crosslinkers by a controlled mixing experiment. Then they show that they can indeed recover known structural changes in a set of purified proteins (complexes) - starting with single subunit proteins up to a very large 0.5 MDa multi-subunit protein complex - the polII complex.The authors give a very measured and fair assessment of this novel isobaric crosslinker and its potential power to contribute to the study of protein structure changes. They show that indeed their novel strategy picks up expected structural changes, changes in surface exposure of certain protein domains, changes within a single protein subunit but also changes in protein-protein interactions. However, they also point out that not all expected dynamic changes are captured and that there is still considerable room for improvement (many not limited to this crosslinker specifically but many crosslinkers used for CSMS).Taken together the study presents a novel set of isobaric crosslinkers that indeed open up the opportunity to provide better qCSMS data, which will enable researchers to study dynamic changes in the shape and structure of proteins and their complexes. However, in its current form, the study some aspects of the study should be expanded upon in order for the research community to assess the true power of these isobaric crosslinkers. Specifically:Although the authors do mention some of the current weaknesses of their isobaric crosslinkers and qCSMS in general, more detail would be extremely helpful. Throughout the article a few key numbers (or even discussions) that would allow one to better evaluate the sensitivity (and the applicability) of the method are missing. This includes:(1) Throughout all the performed experiments it would be helpful to provide information on how many peptides are identified per experiment and how many have actually a crosslinker attached to it.

As the goal of the experiments is to maximize identification of crosslinked peptides which tend to have higher charge states, we targeted ions with charge states of 3+ or higher in our MS acquisition settings for CLMS, and ignored ions with 2+ charge states, which correspond to many of the normal (i.e., not crosslinked) peptides that are identified by MS. As a result, normal peptides are less likely to be identified by the MS procedure used in our CLMS experiments compared to MS settings typically used to identify normal peptides. Our settings may also fail to identify some mono-modified peptides. Like most other CLMS methods, the total number of identified crosslinked peptide spectra is usually less than 1% of the total acquired spectra and we normally expect the crosslinked species to be approximately 1% of the total peptides.

We added information about the number of crosslinked and monolinked peptides identified in the pol I benchmarking experiments (line 173). The number of crosslinks and monolinks identified in the pol II +/- a-amanitin experiment, the TBP/TFIIA/TFIIB experiment and the pol II experiment +/- Rpb4/7 are also provided.

(2) Of all the potential lysines that can be modified - how many are actually modified? Do the authors have an estimate for that? It would be interesting to evaluate in a denatured sample the modification efficiency of the isobaric crosslinker (as an upper limit as here all lysines should be accessible) and then also in a native sample. For example, in the MBP experiment, the authors report the change of one mono-linked peptide in samples containing maltose relative to the one not containing maltose. The authors then give a great description of why this fits to known structural changes. What is missing here is a bit of what changes were expected overall and which ones the authors would have expected to pick up with their method and why have they not been picked up. For example, were they picked up as modified by the crosslinker but not differential? I think this is important to discuss appropriately throughout the manuscript to help the reader evaluate/estimate the potential sensitivity of the method. There are passages where the authors do an excellent job doing that - for example when they mention the missed site that they expected to see in the initial the pol II experiments (lines 191 to 207). This kind of "power analysis" should be heavily discussed throughout the manuscript so that the reader is better informed of what sensitivity can be expected from applying this method.

Regarding the Pol II complex experiment described in Figures 4 and 5, out of the 277 lysine residues in the complex, 207 were identified as monolinked residues (74.7%), and 817 crosslinked pairs out of 38,226 potential pairs (2.1%) were observed. The ability of CLMS to detect proximity/reactivity changes may be impacted by several factors including (1) the (low) abundance of crosslinked peptides in complex mixtures, (2) the presence of crosslinkable residues in close proximity with appropriate orientation, and (3) the ability to generate crosslinked peptides by enzymatic digestion that are amenable to MS analysis (i.e., the peptides have appropriate m/z’s and charge states, the peptides ionize well, the peptides produce sufficient fragment ions during MS2 analysis to allow confident identification). Future efforts to enrich crosslinked peptides prior to MS analysis may improve sensitivity.

It is very difficult to estimate the modification efficiency of Qlinker (or many other crosslinkers) based on peptide identification results. One major reason for this is that trypsin is not able to cleave after a crosslinker-modified lysine residue. As a result, the peptides generated after the modification reaction have different lengths, compositions, charge states, and ionization efficiencies compared to unmodified peptides. These differences make it very difficult to estimate the modification efficiencies based on the presence/absence of certain peptide ions, and/or the intensities of the modified and unmodified versions of a peptide. Also, 2+ ions which correspond to many normal (i.e., unmodified) peptides were excluded by our MS acquisition settings.

It is also very difficult to predict which structural changes are expected and which crosslinked peptides and/or modified peptides can be observed by MS. This is especially true when the experiment involves proteins containing unstructured regions such as the experiments involving Pol II, and TBP, TFIIA and TFIIB. Since we are at the early stages of using qCLMS to study structural changes, we are not sure which changes we can expect to observe by qCLMS. Additional applications of Qlinker-CLMS are needed to better understand the types of structural changes that can be studied using the approach.

We hope that our discussions of some the limitations of CLMS for detecting conformational/reactivity changes provide the reader with an understanding of the sensitivity that can be expected with the approach. At the end of the paragraph about the pol II a-amanitin experiment we say, “Unfortunately, no Q2linker-modified peptides were identified near the site where α-amanitin binds. This experiment also highlights one of the limitations of residue-specific, quantitative CLMS methods in general. Reactive residues must be available near the region of interest, and the modified peptides must be identifiable by mass spectrometry.” In the section about Rbp4/7-induced structural changes in pol II we describe the under-sampling issue. And in the last paragraph we reiterate these limitations and say, “This implies that this strategy, like all MS-based strategies, can only be used for interpretation of positively identified crosslinks or monolinks. Sensitivity and under sampling are common problems for MS analysis of complex samples.”

(3) It would be very helpful to provide information on how much better (or not) the Qlinker approach works relative to label-free qCLMS. One is missing the reference to a potential qCLMS gold standard (data set) or if such a dataset is not readily available, maybe one of the experiments could be performed by label-free qCLMS. For example, one of the differential biosensor experiments would have been well suited.

We agree with the reviewer that it will be very helpful to establish gold standard datasets for CLMS. As we further develop and promote this technology, we will try to establish a standardized qCLMS.

**Reviewer #1 (Recommendations for the authors):**
Only a very minor point:I may have missed it but it's not really clear how many independent experiments were used for the benchmarking quantitation and mixing experiments for Figure 1. What is the reproducibility across experiments on average and on a per-peptide basis?Otherwise, I think the approach would really benefit from at least "Q5linkers" or even "Q10linkers", if possible. And then conduct detailed quantitative studies, either using dilution series or maybe investigating the kinetics of complex formation.

We used a sample of BSA crosslinked peptides to optimize the MS settings, establish the MS acquisition strategies and test the quantification schemes. The data in Figure 1 is based on one experiment, in which used ~150 ug of purified pol I complexes from a 6 L culture. We added this information to the Figure 1 legend. We also provide information about the reproducibility of peptide quantification by plotting the observed and expected ratios for each monolinked and crosslinked peptide identified in all of the runs in Figure S3.

We agree with the reviewer that the Qlinker approach would be even more attractive if multiplex Qlinker reagents were designed. The multiplexed Qlinkers are more difficult and more expensive to synthesize. We are currently working on different schemes for synthesizing multiplexed Qlinkers.

**Reviewer #2 (Recommendations for the authors):**
In addition to the public review I have the following recommendations/questions:(1) The first part of the results section where the synthesis of the crosslinker is explained is excellent for mass spec specialists, but problematic for general readers - either more info should be provided (e.g. b1+ ions - most readers will have no idea why that is) - or potentially it could be simplified here and the details shifted to Materials and Methods for the expert reader. The same is true below for the length of spacer arms.However - in general this level of detail is great - but can impact the ease of understanding for the more mass spec affine but not expert reader.

We have added the following sentence to assist the general reader: A b1+ ion is an ion with a charge state of +1 corresponding to the first N-terminal amino acid residue after breakage of the first peptide bond (lines 126-128).

(2) The Calmodulin experiment (lines 239 to 257) - it is a very nice result that they see the change in the crosslinked peptide between residues K78-K95, but the monolinks are not just detected as described in the text but actually go 2 fold up. This would have been actually a bit expected if the residues are now too far away to be still crosslinked that the monolinks increase. In this case, this counteraction of monolinks to crosslinked sites can also be potentially used as a "selection criteria" for interesting sites that change. Is that a possible interpretation or do the authors think that upregulation of the monolinks is a coincidence and should not be interpreted?

We agree with the reviewer that both monolinks and crosslinks can be used as potential indicators for some changes. However, it is much more difficult to interpret the abundance information from monolinks because, unlike crosslinks, there is little associated structural/proximity information with monolinks. Because it is difficult to understand the reason(s) for changes in monolink abundance, we concentrate on changes in crosslink abundances, which provide proximity/structural information about the crosslinked residues.

(3) Lines 267 to 274: a small thing but the structural information provided is quite dense I have to say. Maybe simplify or accompany with some supplemental figures?

We agree that the structural information is a bit dense especially for readers who are not familiar with the pol II system. We added a reference to Figure 3c (line 177) to help the reader follow the structural information.

As qCLMS is still a relatively new approach for studying conformational changes, the utility of the approach for studying different types of conformational changes is still unclear. Thus, one of the goals of the experiments is to demonstrate the types of conformational changes that can be detected by Q2linkers. We hope that the detailed descriptions will help structural biologists understand the types of conformational changes that can be detected using Qlinkers.

(4) Line 280: explain maybe why the sample was fractionated by SCX (I guess to separate the different complexes?).

SCX was used to reduce the complexity of the peptide mixtures. As the samples are complex and crosslinked peptides are of low abundance compared to normal peptides, SCX can separate the peptides based on their positive charges. Larger peptides and peptides with higher charge states, such as crosslinked peptides, tend to elute at higher salt concentration during SCX chromatography. The use of SCX to fractionate complex peptide mixtures is described in the “General crosslinking protocol and workflow optimization” section of the Methods, and we added a sentence to explain why the sample was fractionated by SCX (lines 278-279).

(5) Lines 354 to 357: "This suggests that the inability to identity most of these crosslinked peptides in both experiments is mainly due to under-sampling during mass spectrometry analysis of the complex samples, rather than the absence of the crosslinked peptides in one of the experiments."This is an extremely important point for the interpretation of missing values - have the authors tried to also collect the mass spec data with DIA which is better in recovery of the same peptide signals between different samples? I realize that these are isobaric samples so DIA measurements per se are not useful as the quantification is done on the reporter channels in the MS2, but it would at least give a better idea if the missing signals were simply not picked up for MS2 as claimed by the authors or the modified peptides are just not present. Another possibility is for the authors to at least try to use a "match between the run" function as can be done in Maxquant. One of the strengths of the method is that it is quantitative and two states are analyzed together, but as can be seen in this experiment, more than two states might want to be compared. In such cases, the under-sampling issue (if that is indeed the cause) makes interpretation of many sites hard (due to missing values) and it would be interesting if for example, an analysis approach with a "match between the runs" function could recover some of the missing values.

We agree that undersampling/missing values is an important issue that needs to be addressed more thoroughly. This also highlights the importance of qCLMS, as conclusions about structural changes based on the presence/absence of certain crosslinked species in database search results may be misleading if the absence of a species is due to under-sampling. We have not tried to collect the data with DIA since we would lose the quantitative information. It would be interesting to see if match between runs can recover some of the missing values. While this could provide evidence to support the under-sampling hypothesis, it would not recover the quantitative information.

We recommend performing label swap experiments and focusing downstream analysis on the crosslinks/monolinks that are identified on both experiments. Future development of multiplexed Qlinker reagents should help to alleviate under-sampling issues. See response to Reviewer #1.

(6) Lines 375 to 393 (the whole paragraph): extremely detailed and not easy to follow. Is that level of detail necessary to drive home that point or could it be visualized in enough detail to help follow the text?

We agree that the paragraph is quite detailed, but we feel that the level of detailed is necessary to describe the types of conformational changes that can be detected by the quantitative crosslinking data, and also illustrate the challenges of interpreting the structural basis for some crosslink abundance changes even when high resolution structural data exists.

To make it easier to follow, we added a sentence to the legend of Figure 5b. “In the holo-pol II structure (right), Switch 5 bending pulls Rpb1:D1442 away from K15, breaking the salt bridge that is formed in the core pol II structure (left). The increase in the abundances of the Rpb1:15-Rpb6:76 and Rpb1:15-Rpb6:72 crosslinks in holo-pol II is likely attributed to the salt bridge between K15 and D1442 in core pol II which impedes the NHS ester-based reaction between the epsilon amino group of K15 and the crosslinker.”

(7) Final paragraph in the results section - lines 397 and 398: "All of the intralinks involving Rpb4 are more abundant in holo-pol II as expected." If I understand that experiment correctly the intralinks with Rpb4 should not be present at all as Rpb4 has been deleted. Is that due to interference between the 126 and 127 channels in MS2? If so, then this also sets a bit of the upper limit of quantitative differences that can be seen. The authors should at least comment on that "limitation".

Yes, we shouldn’t detect any Rpb4 peptides in the sample derived from the Rpb4 knockout strain. The signal from Rpb4 peptides in the DRpb4 sample is likely due to co-eluting ions. To clarify, we changed the text to:

All of the intralinks involving Rpb4 are more abundant in the holo-pol II sample (even though we don’t expect any reporter ion signal from Rpb4 peptides derived from the ∆Rpb4 pol II sample, we still observed reporter ion signals from the channel corresponding to the DRpb4 sample, potentially due to the presence of low abundance, co-eluting ions)(lines 395-399).

(8) Materials and Methods - line 690: I am probably missing something but why were two different mass additions to lysine added to the search (I would have expected only one for the crosslinker)?

The 297 Da modification is for monolinked peptides with one end of the crosslinker hydrolyzed and 18 Da water molecule is added. The 279 Da modification is for crosslinks and sometimes for looplinks (crosslinks involving two lysine residues on the same tryptic peptide).